# Position-based Multiple-play Bandit Problem with Unknown Position Bias

**Junpei Komiyama**
The University of Tokyo
junpei@komiyama.info

**Junya Honda**
The University of Tokyo / RIKEN
honda@stat.t.u-tokyo.ac.jp

**Akiko Takeda**
The Institute of Statistical Mathematics / RIKEN
atakeda@ism.ac.jp

## Abstract

Motivated by online advertising, we study a multiple-play multi-armed bandit problem with position bias that involves several slots and the latter slots yield fewer rewards. We characterize the hardness of the problem by deriving an asymptotic regret bound. We propose the Permutation Minimum Empirical Divergence (PMED) algorithm and derive its asymptotically optimal regret bound. Because of the uncertainty of the position bias, the optimal algorithm for such a problem requires non-convex optimizations that are different from usual partial monitoring and semi-bandit problems. We propose a cutting-plane method and related bi-convex relaxation for these optimizations by using auxiliary variables.

## 1 Introduction

One of the most important industries related to computer science is online advertising. In the United States, 72.5 billion dollars was spent on online advertising [19] in 2016. Most online advertising is viewed on web pages during Internet browsing. A web-site owner has a set of possible advertisements (ads): some of them are more attractive than others, and the owner would like to maximize the attention of visiting users. One of the observable metrics of the user attention is the number of clicks on the ads. By considering each ad (resp. click) to be an arm (resp. reward) and assuming only one slot is available for advertisements, the maximization of clicks boils down to the so-called multi-armed bandit problem, where the arm with the largest expected reward is sought.

When two or more ad slots are available on the web page, the problem boils down to a multiple-play multi-armed bandit problem. Several variants of the multiple play bandit problem and its extension called semi-bandit problem have been considered in the literature. Arguably, the simplest is one assuming that an ad receives equal clicks regardless of its position [2, 24]. In practice, ads receive less clicks when they are placed at bottom slots; this is so-called position bias.

A well-known model that explains position bias is the cascade model [23], which assumes that the users' attention goes from top to bottom until they lose interest. While this model explains position bias in early positions well [10], a drawback to the cascade model when it is applied to the bandit setting [26] is that the order of the allocated ads does not affect the reward, which is not very natural. To resolve this issue, Combes et al. [8] introduced a weight for each slot that corresponds to the reward obtained by clicking on that slot. However, no principled way of defining the weight has been described.

An extension of the cascade model, called the dependent click model (DCM) [14], addresses these issues by admitting multiple clicks of a user. In DCM, each slot is associated with a probability that

the user loses interest in the following ads if the current ad is interesting. While the algorithm in Katariya et al. [21] cleverly exploits this structure, it still depends on the cascade assumption, and as a result it discards some of the feedback on the latter slots, which reduces the efficiency of the algorithm. Moreover, the reward in DCM does not exactly correspond to the number of clicks.

Lagrée et al. [27] has studied a position-based model (PBM) where each slot has its own discount factor on the number of clicks. PBM takes the order of the shown ads into consideration. However, the algorithms proposed in Lagrée et al. [27] are "half-online" in the sense that the value of an ad is adaptively estimated, whereas the values of the slots are estimated by using an off-line dataset. Such an off-line computation is not very handy since the click trend varies depending on the day and hour [1]. Moreover, a significant portion of online advertisements is sold via ad networks [34]. As a result, advertisers have to deal with thousands of web pages to show their ads. Taking these aspects into consideration, pre-computing position bias for each web page limits the use of these algorithms.

To address this issue, we provide a way to allocate advertisements in a fully online manner by considering "PBM under Uncertainty of position bias" (PBMU). One of the challenges when the uncertainty of a position-based factor is taken into account is that, when some ad appears to have a small click through rate (CTR, the probability of click) in some slot, we cannot directly attribute it to either the arm or the slot. In this sense, several combinations of ads and slots need to be examined to estimate both the ad-based and position-based model parameters.

Note also that an extension of the non-stochastic bandit approach [3] to multiple-play, such as the ordered slate model [20], is general enough to deal with PBMU. However, algorithms based on the non-stochastic approach do not always perform well in compensation for its generality. Another extension of multi-armed bandit problems is the partial monitoring problem [31, 4] that admits the case in which the parameters are not directly observable. However, partial monitoring is inefficient at solving bandit problems: a $K$-armed bandit problem with binary rewards corresponds to a partial monitoring problem with $2^K$ possible outcomes. As a result, the existing partial monitoring algorithms, such as the ones in [33, 25], are not practical even for a moderate number of arms. Besides, the computation of a feasible solution in PBMU requires non-convex optimizations as we will see in Section 5. This implies that PBMU cannot directly be converted into the partial monitoring where such a non-convex optimization does not appear [25].

The contributions of this paper are as follows: First, we study the position-based bandit model with uncertainty (PBMU) and derive a regret lower bound (Section 3). Second, we propose an algorithm that efficiently utilizes feedback (Section 4). One of the challenges in the multiple-play bandit problem is that there is an exponentially large number of possible sequences of arms to allocate at each round. We reduce the number of candidates by using a bipartite matching algorithm that runs in a polynomial time to the number of arms. The performance of the proposed algorithm is verified in Section 6. Third, a slightly modified version of the algorithm is analyzed in Section 7. This algorithm has a regret upper bound that matches the lower bound. Finally, we reveal that the lower bound is related to a linear optimization problem with an infinite number of constraints. Such an optimization problem appears in many versions of the bandit problem [9, 25, 12]. We propose an optimization method that reduces it to a finite-constraint linear optimization based on a version of the cutting-plane method (Section 5). Related non-convex optimizations that are characteristic to PBMU are solved by using bi-convex relaxation. Such optimization methods are of interest in solving even larger classes of bandit problems.

## 2 Problem Setup

Let $K$ be the number of arms (ads) and $L < K$ be the number of slots. Each arm $i \in [K] = \{1, 2, \ldots, K\}$ is associated with a distinct parameter $\theta_i^* \in (0, 1)$, and each slot $l \in [L]$ is associated with a parameter $\kappa_l^* \in (0, 1]$. At each round $t = 1, 2, \ldots, T$, the system selects $L$ arms $I(t) = (I_1(t), \ldots, I_L(t))$ and receives a corresponding binary reward (click or non-click) for each slot. The reward of the $l$-th slot is i.i.d. drawn from a Bernoulli distribution $\mathrm{Ber}(\mu_{I_l(t),l}^*)$, where $\mu_{i,l}^* = \theta_i^* \kappa_l^*$. Although the slot-based parameters are unknown, it is natural that the ads receives more clicks when they are placed at early slots: we assume $\kappa_1^* > \kappa_2^* > \cdots > \kappa_L^* > 0$ and this order is known.

Note that this model is redundant: a model with $\mu_{i,l}^* = \theta_i^* \kappa_l^*$ is equivalent to the model with $\mu_{i,l}^* = (\theta_i^*/\kappa_1)(\kappa_l^* \kappa_1)$. Therefore, without loss of generality, we assume $\kappa_1 = 1$. In summary,

this model involves $K + L$ parameters $\{\theta_i^*\}_{i \in [K]}$ and $\{\kappa_l^*\}_{l \in [L]}$, and the number of rounds $T$. The parameters except for $\kappa_1 = 1$ are unknown to the system. Let $N_{i,l}(t)$ be the number of rounds before $t$-th round at which arm $i$ was in slot $l$ (i.e., $N_{i,l}(t) = \sum_{t'=1}^{t-1} \mathbf{1}\{i = I_l(t')\}$, where $\mathbf{1}\{\mathcal{E}\}$ is 1 if $\mathcal{E}$ holds and 0 otherwise). In the following, we abbreviate arm $i$ in slot $l$ to "pair $(i,l)$". Let $\hat{\mu}_{i,l}(t)$ be the empirical mean of the reward of pair $(i,l)$ after the first $t-1$ rounds.

The goal of the system is to maximize the cumulative rewards by using some sophisticated algorithm. Without loss of generality, we can assume $\theta_1^* > \theta_2^* > \theta_3^* > \cdots > \theta_K^*$. The algorithm cannot exploit this ordering. In this model, allocating arms of larger expected rewards on earlier slots increases expected rewards: As a result, allocating arms $1, 2, \ldots, L$ to slots $1, 2, \ldots, L$ maximizes the expected reward. A quantity called (pseudo) regret is defined as: $\mathrm{Reg}(T) = \sum_{t=1}^{T} \left( \sum_{i \in [L]} (\theta_i^* - \theta_{I_i(t)}^*) \kappa_i^* \right)$, and $\mathbb{E}[\mathrm{Reg}(T)]$ is used for evaluating the performance of an algorithm. Let $\Delta_{i,l} = \theta_l^* \kappa_l^* - \theta_i^* \kappa_l^*$. Regret can be alternatively represented as $\mathrm{Reg}(T) = \sum_{(i,l) \in [K] \times [L]} \Delta_{i,l} N_{i,l}(T)$. The regret increases unless $I(t) = (1, 2, \ldots, L)$.

## 3 Regret Lower Bound

Here, we derive an asymptotic regret lower bound when $T \to \infty$. In the context of the standard multi-armed bandit problem, Lai and Robbins [28] derived a regret lower bound for strongly consistent algorithms, and it is followed by many extensions, such as the one for multi-parameter distributions [6] and the ones for Markov decision processes [13, 7]. Intuitively, a strongly consistent algorithm is "uniformly good" in the sense that it works well with any set of model parameters. Their result was extended to the multiple-play [2] and PBM [27] cases. We further extend it to the case of PBMU.

Let $\mathcal{T}_{\mathrm{all}} = \{(\theta_1', \ldots, \theta_K') \in (0,1)^K\}$ and $\mathcal{K}_{\mathrm{all}} = \{(\kappa_1', \ldots, \kappa_L') : 1 = \kappa_1' > \kappa_2' > \cdots > \kappa_L' > 0\}$ be the sets of all possible values on the parameters of the arms and slots, respectively. Let $(1), \ldots, (K)$ be a permutation of $1, \ldots, K$ and $\mathcal{T}_{(1),\ldots,(L)}$ be the subset of $\mathcal{T}_{\mathrm{all}}$ such that the $i$-th best arm is $(i)$. Namely,

$$\mathcal{T}_{(1),\ldots,(L)} = \left\{ (\theta_1', \ldots, \theta_K') \in (0,1)^K : \theta_{(1)}' > \theta_{(2)}' > \cdots > \theta_{(L)}', \forall_{i \notin \{(1),\ldots,(L)\}} (\theta_i' < \theta_{(L)}') \right\},$$

and $\mathcal{T}_{(1),\ldots,(L)}^c = \mathcal{T}_{\mathrm{all}} \setminus \mathcal{T}_{(1),\ldots,(L)}$. An algorithm is *strongly consistent* if $\mathbb{E}[\mathrm{Reg}(T)] = o(T^a)$ for any $a > 0$ given any instance of the bandit problem with its parameters $\{\theta_i'\}_{i \in [K]} \in \mathcal{T}_{\mathrm{all}}$, $\{\kappa_l'\} \in \mathcal{K}_{\mathrm{all}}$. The following lemma, whose proof is in Appendix F, lower-bounds the number of draws on the pairs of arms and slots.

**Lemma 1.** (Lower bound on the number of draws) *The following inequality holds for $N_{i,l}(T)$ of the strongly consistent algorithm:*

$$\forall_{\{\theta_i'\} \in \mathcal{T}_{1,\ldots,L}^c, \{\kappa_l'\} \in \mathcal{K}_{\mathrm{all}}} \sum_{(i,l) \in [K] \times [L]} \mathbb{E}[N_{i,l}(T)] d_{\mathrm{KL}}(\theta_i^* \kappa_l^*, \theta_i' \kappa_l') \geq \log T - o(\log T),$$

*where $d_{\mathrm{KL}}(p,q) = p \log(p/q) + (1-p) \log((1-p)/(1-q))$ is the KL divergence between two Bernoulli distributions.*

Such a divergence-based bound appears in many stochastic bandit problems. However, unlike other bandit problems, the argument inside the KL divergence is a product of parameters $\theta_i' \kappa_l'$: While $d_{\mathrm{KL}}(\cdot, \theta_i' \kappa_l')$ is convex to $\theta_i' \kappa_l'$, it is not convex to the parameter space $\{\theta_i'\}, \{\kappa_l'\}$. Therefore, finding a set of parameters that minimizes $\sum_{i,l} d_{\mathrm{KL}}(\mu_{i,l}, \theta_i' \kappa_l')$ is non-convex, which makes PBMU difficult.

Furthermore, we can formalize the regret lower bound in what follows. Let

$$\mathcal{Q} = \left\{ \{q_{i,l}\} \in [0,\infty)^{[K] \times [K]} : \forall_{i \in [K-1]} \sum_{l \in [K]} q_{i,l} = \sum_{l \in [K]} q_{i+1,l}, \forall_{l \in [K-1]} \sum_{i \in [K]} q_{i,l} = \sum_{i \in [K]} q_{i,l+1} \right\}.$$

Intuitively, $\{q_{i,l}\}$ for $l \leq L$ corresponds to the draw of arm $i$ in slot $l$, and $\{q_{i,l}\}$ for $l > L$ corresponds to the non-draw of arm $i$, as we will see later. The following quantities characterizes the minimum

amount of exploration for consistency:

$$\mathcal{R}_{(1),\ldots,(L)}(\{\mu_{i,l}\}, \{\theta_i\}, \{\kappa_l\}) = \left\{ \{q_{i,l}\} \in \mathcal{Q} : \inf_{\{\theta_i'\} \in \mathcal{T}_{(1),\ldots,(L)}^c, \{\kappa_l'\} \in \mathcal{K}_{\mathrm{all}} : \forall_{i \in [L]} \theta_i' \kappa_i' = \theta_i \kappa_i} \right.$$

$$\left. \sum_{(i,l) \in [K] \times [L] : i \neq (l)} q_{i,l} d_{\mathrm{KL}}\left(\mu_{i,l}, \theta_i' \kappa_l'\right) \geq 1 \right\}. \quad (1)$$

Equality (1) states that drawing each pair $(i, l)$ for $N_{i,l} = q_{i,l} \log T$ times suffices to reduce the risk that the true parameter is $\{\theta_i'\}, \{\kappa_l'\}$ for any parameters $\{\theta_i'\}, \{\kappa_l'\}$ such that $\theta_i' \in \mathcal{T}_{(1),\ldots,(L)}^c$ and $\theta_i' \kappa_l' = \theta_i \kappa_i$ for any $i \in [L]$. Note that the constraint $\theta_i' \kappa_l' = \theta_i \kappa_i$ corresponds to the fact that drawing an optimal list of arms does not increase the regret: Intuitively, this corresponds to the fact that the true parameter of the best arm is obtained for free in the regret lower bound of the standard bandit problem[1]. Moreover, let

$$C_{(1),\ldots,(L)}^*(\{\mu_{i,l}\}, \{\theta_i\}, \{\kappa_l\}) = \inf_{\{q_{i,l}\} \in \mathcal{R}_{(1),\ldots,(L)}(\{\mu_{i,l}\}, \{\theta_i\}, \{\kappa_l\})} \sum_{(i,l) \in [K] \times [L]} \Delta_{i,l} q_{i,l},$$

the set of optimal solutions of which is denoted by

$$\mathcal{R}_{(1),\ldots,(L)}^*(\{\mu_{i,l}\}, \{\theta_i\}, \{\kappa_l\}) = \left\{ \{q_{i,l}\} \in \mathcal{R}_{(1),\ldots,(L)}(\{\mu_{i,l}\}, \{\theta_i\}, \{\kappa_l\}) : \right.$$

$$\left. \sum_{(i,l) \in [K] \times [L]} \Delta_{i,l} q_{i,l} = C_{(1),\ldots,(L)}^*(\{\mu_{i,l}\}, \{\theta_i\}, \{\kappa_l\}) \right\}. \quad (2)$$

The value $C_{1,\ldots,L}^* \log T$ is the possible minimum regret such that the minimum divergence of $\{\theta_i^*\}, \{\kappa_l^*\}$ from any $\{\theta_i'\}, \{\kappa_l'\}$ is larger than $\log T$. Using Lemma 1 yields the following regret lower bound, whose proof is also in the Appendix F.

**Theorem 2.** *The regret of a strongly consistent algorithm is lower bounded as follows:*

$$\mathbb{E}[\mathrm{Reg}(T)] \geq C_{1,\ldots,L}^*(\{\mu_{i,l}^*\}, \{\theta_i^*\}, \{\kappa_l^*\}) \log T - o(\log T).$$

**Remark 3.** $N_{i,l} = (\log T)/d_{\mathrm{KL}}(\theta_i^* \kappa_i^*, \theta_j^* \kappa_i^*)$ for $j = \min(i - 1, L)$ satisfies the conditions in Lemma 1, which means that regret lower bound in Theorem 2 is $O(K \log T / \Delta) = O(K \log T)$, where $\Delta = \min_{i \neq j, l \neq m} |\theta_i^* - \theta_j^*| |\kappa_l^* - \kappa_m^*|$.

## 4 Algorithm

Our algorithm, called Permutation Minimum Empirical Divergence (PMED), is closely related to the optimization we discussed in Section 3.

### 4.1 PMED Algorithm

We denote a list of $L$ arms that are drawn at each round as $L$-allocation. For example, $(3, 2, 1, 5)$ is a 4-allocation, which corresponds to allocating arms $3, 2, 1, 5$ to slots $1, 2, 3, 4$, respectively. Like the Deterministic Minimum Empirical Divergence (DMED) algorithm [17] for the single-play multi-armed bandit problem, Algorithm 1 selects arms by using a loop. $L_C = L_C(t)$ is the set of $L$-allocations in the current loop, and $L_N = L_N(t)$ is the set of $L$-allocations that are to be drawn in the next loop. Note that, $|L_N| \geq 1$ always holds at the end of each loop so that at least one element is

**Algorithm 1** PMED and PMED-Hinge Algorithms
---
1: Input: $\alpha > 0$, $\beta > 0$ (for PMED-Hinge), $f(n) = \gamma/\sqrt{n}$ with $\gamma > 0$ (for PMED-Hinge).
2: $L_N \leftarrow \emptyset$. $L_C \leftarrow \{v_1^{\mathrm{mod}}, \ldots, v_K^{\mathrm{mod}}\}$.
3: **while** $t \leq T$ **do**
4:    **for** each $v_m^{\mathrm{mod}} : m \in [K]$ **do**
5:       If there exists some pair $(i,l) \in v_m^{\mathrm{mod}}$ such that $N_{i,l}(t) < \alpha\sqrt{\log t}$, then put $v_m^{\mathrm{mod}}$ into $L_N$.
6:    **end for**
7:    Compute the MLE $\{\hat{\theta}_i(t)\}_{i=1}^K$, $\{\hat{\kappa}_l(t)\}_{l=1}^L$

$$= \begin{cases} \min_{\{\theta_i, \kappa_l\}} \sum_{(i,l)\in[K]\times[L]} N_{i,l}(t) d_{\mathrm{KL}}(\hat{\mu}_{i,l}(t), \theta_i \kappa_l) & \text{(PMED)} \\ \min_{\{\theta_i, \kappa_l\}} \sum_{(i,l)\in[K]\times[L]} N_{i,l}(t) \left( d_{\mathrm{KL}}(\hat{\mu}_{i,l}(t), \theta_i \kappa_l) - f(N_{i,l}(t)) \right)_+ . & \text{(PMED-Hinge)} \end{cases}$$

8:    **if** Algorithm is PMED-Hinge **then**
9:       If $|\hat{\theta}_i(t) - \hat{\theta}_j(t)| < \beta/(\log\log t)$ for some $i \neq j$ or $|\hat{\kappa}_l(t) - \hat{\kappa}_m(t)| < \beta/(\log\log t)$ for some $l \neq m$, then put all of $v_1^{\mathrm{mod}}, \ldots, v_K^{\mathrm{mod}}$ to $L_N$.
10:       If $\bigcup_{(i,l)\in[K]\times[L]}\{d_{\mathrm{KL}}(\hat{\mu}_{i,l}(t), \hat{\theta}_i(t)\hat{\kappa}_l(t)) > f(N_{i,l}(t))\}$ holds, then put all of $v_1^{\mathrm{mod}}, \ldots, v_K^{\mathrm{mod}}$ into $L_N$.
11:    **end if**
12:    Compute $\{q_{i,l}\} \in \begin{cases} \mathcal{R}^*_{\hat{1}(t),\ldots,\hat{L}(t)}(\{\hat{\mu}_{i,l}(t)\}, \{\hat{\theta}_i(t)\}, \{\hat{\kappa}_l(t)\}) & \text{(PMED)} \\ \mathcal{R}^{*,\mathrm{H}}_{\hat{1}(t),\ldots,\hat{L}(t)}(\{\hat{\mu}_{i,l}(t)\}, \{\hat{\theta}_i(t)\}, \{\hat{\kappa}_l(t)\}, \{f(N_{i,l}(t))\}). & \text{(PMED-Hinge)} \end{cases}$
13:    $\tilde{N}_{i,l} \leftarrow q_{i,l} \log t$ for each $(i,l) \in [K] \times [K]$.
14:    Decompose $\tilde{N}_{i,l} = \sum_v c_v^{\mathrm{req}} e_v$ where $e_v$ for each $v$ is a permutation matrix and $c_v^{\mathrm{req}} > 0$ by using Algorithm 2.
15:    $r_{i,l} \leftarrow N_{i,l}(t)$.
16:    **for** each permutation matrix $e_v$ **do**
17:       $c_v^{\mathrm{aff}} \leftarrow \min\left(c_v^{\mathrm{req}}, \max_c \{c > 0 : \min_{(i,l)\in[K]\times[L]}(r_{i,l} - c\, e_{v,i,l}) \geq 0\}\right)$.
18:       Let $(v_1, \ldots, v_L)$ be the $L$-allocation corresponding to $e_v$. If $c_v^{\mathrm{aff}} < c_v^{\mathrm{req}}$ and there exists a pair $(v_l, l)$ that is in none of the $L$-allocations in $L_N$, then put $(v_1, \ldots, v_L)$ into $L_N$.
19:       $r_{i,l} \leftarrow r_{i,l} - c_v^{\mathrm{aff}} e_{v,i,l}$.
20:    **end for**
21:    Select $I(t) \in L_C$ in an arbitrary fixed order. $L_C \rightarrow L_C \setminus \{I(t)\}$.
22:    Put $(\hat{1}(t), \ldots, \hat{L}(t))$ into $L_N$.
23:    If $L_C = \emptyset$ then $L_C \leftarrow L_N$, $L_N \leftarrow \emptyset$.
24: **end while**
---

put into $L_C$. There are three lines where $L$-allocations are put into $L_N$ without duplication: Lines 5, 18, and 22. We explain each of these lines below.

Line 5 is a uniform exploration over all pairs $(i,l)$. For $m \in [K]$, let $v_m^{\mathrm{mod}}$ be an $L$-allocation $(1 + \mathrm{mod}_K(m), 1 + \mathrm{mod}_K(1+m), \ldots, 1 + \mathrm{mod}_K(L+m-1))$, where $\mathrm{mod}_K(x)$ is the minimum non-negative integer among $\{x - cK : c \in \mathbb{N}\}$. From the definition of $v_m^{\mathrm{mod}}$, any pair $(i,l) \in [K]\times[L]$ belongs to exactly one of $v_1^{\mathrm{mod}}, \ldots, v_K^{\mathrm{mod}}$. If some pair $(i,l)$ is not allocated $\alpha\sqrt{\log t}$ times, a corresponding $L$-allocation is put into $L_N$. This exploration stabilizes the estimators.

Line 18 and related routines are based on the optimal amount of explorations. $\{\tilde{N}_{i,l}\}_{i\in[K],l\in[K]}$ is calculated by plugging in the maximum likelihood estimator (MLE) $(\{\hat{\theta}_i\}_{i\in[K]}, \{\hat{\kappa}_l\}_{l\in[L]})$ into the optimization problem of Inequality (2). As $\{\tilde{N}_{i,l}\}$ is a set of $K \times K$ variables[2], the algorithm needs to convert it into a set of $L$-allocations to put them into $L_N$. This is done by decomposing it into a set of permutation matrices, which we will explain in Section 4.2.

Line 22 is for exploitation: If no pair is put to $L_N$ by Line 5 or Line 18 and $L_C$ is empty, then Line 22 puts arms $(\hat{1}(t), \ldots, \hat{L}(t))$ of the top-$L$ largest $\{\hat{\theta}_i(t)\}$ (with ties broken arbitrarily) into $L_N$.

**Algorithm 2** Permutation Matrix Decomposition

---

1: Input: $N_{i,l}$.
2: $\bar{N}_{i,l} \leftarrow N_{i,l}$.
3: **while** $\bar{N}_{i,l} > 0$ for some $(i,l) \in [K] \times [K]$ **do**
4:     Find a permutation matrix $e_v$ such that, for any $i,l$ such that $e_{v,i,l} = 1 \Rightarrow \bar{N}_{i,l} > 0$.
5:     Let $c_v^{\text{req}} = \max_c \left\{ c > 0 : \min_{(i,l) \in [K] \times [K]} (\bar{N}_{i,l} - c e_{v,i,l}) \geq 0 \right\}$.
6:     $\bar{N}_{i,l} \leftarrow \bar{N}_{i,l} - c_v^{\text{req}} e_{v,i,l}$ for each $(i,l) \in [K] \times [K]$.
7: **end while**
8: Output $\{c_v^{\text{req}}, e_v\}$

---

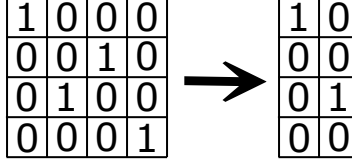

Figure 1: A permutation matrix with $K = 4$, where $(i,l) = 1$ for $(i,l) \in (1,1), (2,3), (3,2), (4,4)$. If $L = 2$, this matrix corresponds to allocating arm 1 in slot 1 and arm 3 in slot 2.

## 4.2 Permutation Matrix and Allocation Strategy

In this section, we discuss the way to convert $\{\tilde{N}_{i,l}\} = \{q_{i,l} \log t\}$, the estimated optimal amount of exploration, into $L$-allocations. A permutation matrix is a square matrix that has exactly one entry of 1 at each row and each column and 0s elsewhere (Figure 1, left). There are $K!$ permutation matrices since they corresponds to ordering $K$ elements. Therefore, even though $\{q_{i,l}\}$ can be obviously decomposed into a linear combination of permutation matrices, it is not clear how to compute them without computing the set of all permutation matrices that are exponentially large in $K$. Algorithm 2 solves this problem: Let $\bar{N}_{i,l}$ be a temporal variable that is initialized by $\tilde{N}_{i,l}$ at the beginning. In each iteration, it subtracts a scalar multiplication of a permutation matrix $e_v$ whose $(i,l)$ entry $e_{v,i,l}$ of value 1 corresponds to $\bar{N}_{i,l} > 0$. (Line 6 in Algorithm 2). This boils down to finding a perfect matching in a bipartite graph where the left (resp. right) nodes correspond to rows (resp. columns) and edges between nodes $i$ and $l$ are spanned if $\bar{N}_{i,l} > 0$. Although a naive greedy fails in such a matching problem (c.f., Appendix A), a maximal matching in a bipartite graph can be computed by the Hopcroft–Karp algorithm [18] in $O(K^{2.5})$ times, and Theorem 4 below ensures that the maximum matching is always perfect:

**Theorem 4.** (Existence of a perfect matching) *For any* $\{\bar{N}_{i,l} \in [K] \times [K] : \bar{N}_{i,l} \geq 0, \exists_{(i,l)} \bar{N}_{i,l} > 0\}$ *such that the sums of each row and column are equal, there exists a permutation matrix* $e_v$ *such that* $\forall_{(i,l) \in [K] \times [K] : e_{v,i,l} = 1} \bar{N}_{i,l} > 0$.

The proof of Theorem 4 is in Appendix E. Each subtraction increases the number of 0 entries in $\bar{N}_{i,l}$ (Line 5 in Algorithm 2); Algorithm 2 runs in $O(K^{4.5})$ times by computing at most $O(K^2)$ perfect matching sub-problems, and as a result it decomposes $\tilde{N}_{i,l}$ into a positive linear combination of permutation matrices. The main algorithm checks whether each the entries of the permutation matrices are sufficiently explored (Line 18 in Algorithm 1), and draws an $L$-allocation corresponding to a permutation matrix (Figure 1, right) if under-explored.

## 5 Optimizations

This section discusses two optimizations that appear in Algorithm 1, namely, the MLE computation (Line 7), and the computation of the optimal solution (Line 12).

MLE (Line 7) is the solution of a bi-convex optimization: the optimization of $\{\theta_i\}$ (resp. $\{\kappa_l\}$) is convex when we view $\{\kappa_l\}$ (resp. $\{\theta_i\}$) as a constant. Therefore, off-the-shelf tools for optimizing convex functions (e.g., Newton's method) are applicable to alternately optimizing $\{\theta_i\}$ and $\{\kappa_l\}$. Assuming that each convex optimization yields an optimal value, such an alternate optimization

---

**Algorithm 3** Cutting-plane method for obtaining $\{q_{i,l}\}$ on Line 12 of Algorithm 1

---

1: Input: the number of iterations $S$, nominal constraint $\{\theta_i^{(0)}\} \in \mathcal{T}_{\hat{1}(t),\dots,\hat{L}(t)}^c$.
2: **for** $s = 1, 2, \dots, S$ **do**
3:     Find $q_{i,l}^{(s)} \leftarrow \min_{\{q_{i,l}\} \in \mathcal{Q}} \sum_{(i,l) \in [K] \times [L]} \Delta_{i,l} q_{i,l}$ such that

$$\sum_{(i,l) \in [K] \times [L]:i \neq \hat{l}(t)} q_{i,l} d_{\mathrm{KL}} \left( \hat{\mu}_{i,l}(t), \theta_i' \frac{\hat{\theta}_l(t) \hat{\kappa}_l(t)}{\theta_l'} \right) \geq 1$$

      for all $\{\theta_i'\} \in \{\theta_i^{(0)}\}, \{\theta_i^{(1)}\}, \dots, \{\theta_i^{(s-1)}\}$.
4:     Find $\{\theta_i^{(s)}\} \leftarrow \min_{\{\theta_i'\}} \sum_{(i,l) \in [K] \times [L]} q_{i,l}^{(s)} d_{\mathrm{KL}}(\hat{\mu}_{i,l}(t), \theta_i' \frac{\hat{\theta}_l(t) \hat{\kappa}_l(t)}{\theta_l'})$.
5: **end for**

---

monotonically decreases the objective function and thus converges. Note that a local minimum obtained by bi-convex optimizations is not always a global minimum due to its non-convex nature.

Although the computation of the optimal solution (Line 12) involves $\{\theta_i'\}$ and $\{\kappa_l'\}$, the constraint eliminates latter variables as $\kappa_i' = \hat{\theta}_i(t) \hat{\kappa}_i(t) / \theta_i'$. This optimization is a linear semi-infinite programming (LSIP) on $\{q_{i,l}\}$, which is a linear programming (LP) with an infinite set of linear constraints parameterized by $\{\theta_i'\}$. Algorithm 3 is the cutting-plane method with pessimistic oracle [29] that boils the LSIP down to finite constraint LPs. At each iteration $s$, it adds a new constraint $\{\theta_i^{(s)}\} \in \mathcal{T}_{\hat{1}(t),\dots,\hat{L}(t)}^c$ that is "hardest" in a sense that it minimizes the sum of divergences (Line 4 in Algorithm 3). The following theorem guarantees the convergence of the algorithm when the exactly hardest constraint is found.

**Theorem 5.** (Convergence of the cutting-plane method, Mutapcic and Boyd [29, Section 5.2]) *Assume that there exists a constant $C$ and that the constraint $f(\{\theta_i'\}) = \sum_{(i,l) \in [K] \times [L]} q_{i,l}^{(s)} d_{\mathrm{KL}}(\hat{\mu}_{i,l}(t), \theta_i' \frac{\hat{\theta}_l(t) \hat{\kappa}_l(t)}{\theta_l'})$ is Lipchitz continuous as $|f(\{\theta_i^{(1)}\}) - f(\{\theta_i^{(2)}\})| \leq C||\{\theta_i^{(1)}\} - \{\theta_i^{(2)}\}||$, where the norm $||\cdot||$ is any $L_p$ norm. Then, Algorithm 3 converges to its optimal solution as $S \to \infty$.*

Although the Lipchitz continuity assumption does not hold as $d_{\mathrm{KL}}(p, q)$ approaches infinity when $q$ is close to 0 or 1, by restricting $q$ to some region $[\epsilon, 1 - \epsilon]$, Lipchitz continuity can be guaranteed for some $C = C(\epsilon)$. Theorem 5 assumes the availability of an exact solution to the hardest constraint, which is generally hard since this objective is non-convex in its nature. Still, we can obtain a fair solution with the following reasons: First, although the space $\mathcal{T}_{\hat{1}(t),\dots,\hat{L}(t)}^c$ is not convex, it suffices to consider each of the convex subspaces $\left\{ \{\theta_i'\} \in (0,1)^K : \theta_{\hat{1}(t)}' \geq \dots \geq \theta_{\hat{L}(t)}', \theta_{\hat{X}(t)}' = \theta_{\hat{l}(t)}' \right\}$ where $X = \min(L, l-1)$, for each $l \in [K] \setminus \{1\}$ separately because the hardest constraint is always in one of these subspaces (which follows from the convexity of the objective function). Second, the following bi-convex relaxation can be used: Let $\eta_1', \dots, \eta_L'$ be auxiliary variables that correspond to $1/\theta_1', \dots, 1/\theta_L'$. Namely, we optimize a relaxed objective function $\sum_{(i,l) \in [K] \times [L]} \left( q_{i,l}^{(s)} d_{\mathrm{KL}}(\hat{\mu}_{i,l}(t), \theta_i' \eta_l' \hat{\theta}_l(t) \hat{\kappa}_l(t)) \right) + \phi \sum_{i \in [L]} (\theta_i' \eta_i' - 1)^2$, where $\phi > 0$ is a penalty parameter. Convexity of KL divergence implies that this objective is a bi-convex function of $\{\theta_i'\}$ and $\{\eta_l'\}$, and thus an alternate optimization is effective. Setting $\phi \to \infty$ induces a solution in which $\eta_i'$ is equal to $1/\theta_i'$ ([30, Theorem 17.1]). Our algorithm starts with a small value of $\phi$; then it gradually increases $\phi$.

## 6 Experiment

To evaluate the empirical performance of the proposed algorithms, we conducted computer simulations with synthetic and real-world datasets. The compared algorithms are MP-TS [24], dcmKL-UCB [21], PBM-PIE [27], and PMED (proposed in this paper). MP-TS is an algorithm based on Thompson sampling [32] that ignores position bias: it draws the top-$L$ arms on the basis of posterior sampling, and the posterior is calculated without considering position bias. DcmKL-UCB is a KL-UCB [11]

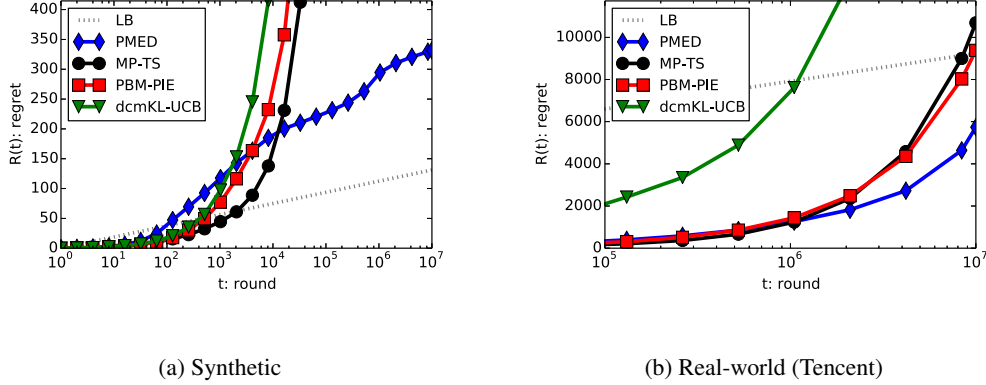

(a) Synthetic  (b) Real-world (Tencent)

Figure 2: Regret-round log-log plots of algorithms.

based algorithm that works under the DCM assumption. PBM-PIE is an algorithm that allocates top-$(L-1)$ slots greedily and allocates $L$-th arm based on the KL-UCB bound. Note that PBM-PIE requires an estimation of $\{\kappa_l^*\}$; here, a bi-convex optimization is used to estimate it[3]. We did not test PBM-TS [27], which is another algorithm for PBM, mainly because that its regret bound has not been derived yet. However, its regret appears to be asymptotically optimal when $\{\kappa_l^*\}$ are known (Figure 1(a) in Lagrée et al.[27]), and thus it does not explore sufficiently when there is uncertainty in the position bias. We set $\alpha = 10$ for PMED. We used the Gurobi LP solver[4] for solving the LPs. To speed up the computation, we skipped the bi-convex and LP optimizations in most rounds with large $t$ and used the result of the last computation. We used the Newton's method (resp. a gradient method) for computing the MLE (resp. the hardest constraint) in Algorithm 3.

**Synthetic data**: This simulation was designed to check the consistency of the algorithms, and it involved 5 arms with $(\theta_1, \ldots, \theta_5) = (0.95, 0.8, 0.65, 0.5, 0.35)$, and 2 slots with $(\kappa_1, \kappa_2) = (1, 0.6)$. The experimental results are shown on the left of Figure 2. The results are averaged over 100 runs. LB is the simulated value of the regret lower bound in Section 3. While the regret of PMED converges, the other algorithms suffer a 100 times or larger regret than LB at $T = 10^7$, which implies that these algorithms are not consistent under our model.

**Real-world data**: Following the existing work [24, 27], we used the KDD Cup 2012 track 2 dataset [22] that involves session logs of soso.com, a search engine owned by Tencent. Each of the 150M lines from the log contains the user ID, the query, an ad, and a slot in $\{1, 2, 3\}$ at which the ad was displayed and a binary reward indicated (click/no-click). Following Lagrée et al. [27], we obtained major 8 queries. Using the click logs of the queries, the CTRs and position bias were estimated in order to maximize the likelihood by using bi-convex optimization in Section 4. Note that, the number of arms and parameters are slightly different from the ones reported previously [27]. For the sake of completeness, we show the parameters in Appendix C. We conducted 100 runs for each queries, and the right figure in Figure 2 shows the averaged regret over 8 queries. Although the gap between PMED and existing algorithms are not drastic compared with synthetic parameters, the existing algorithms suffer larger regret than PMED.

## 7 Analysis

Although the authors conjecture that PMED is optimal, it is hard to analyze it directly. The technically hardest part arises from the case in which the divergence of each action is small but not yet fully converged. To circumvent these difficulty, we devised a modified algorithm called PMED-Hinge (Algorithm 1) that involves extra exploration. In particular, we modify the optimization problem as

follows: Let

$$\mathcal{R}^{\mathrm{H}}_{(1),\dots,(L)}(\{\mu_{i,l}\},\{\theta_i\},\{\kappa_l\},\{\delta_{i,l}\}) = \Bigg\{\{q_{i,l}\} \in \mathcal{Q} : \inf_{\{\theta'_i\}\in\mathcal{T}^c_{(1),\dots,(L)},\{\kappa'_l\}\in\mathcal{K}_{\mathrm{all}}:\forall_{l\in[L]}d_{\mathrm{KL}}(\mu_{(l),l},\theta'_{(l)}\kappa'_l)\le\delta_{i,l}}$$

$$\sum_{(i,l)\in[K]\times[L]:i\neq(l)} q_{i,l}\left(d_{\mathrm{KL}}(\mu_{i,l},\theta'_i\kappa'_l)-\delta_{i,l}\right)_+ \ge 1\Bigg\},$$

where $(x)_+ = \max(x,0)$. Moreover, let

$$C^{*,\mathrm{H}}_{(1),\dots,(L)}(\{\mu_{i,l}\},\{\theta_i\},\{\kappa_l\},\{\delta_{i,l}\}) = \inf_{\{q_{i,l}\}\in\mathcal{R}^{\mathrm{H}}_{(1),\dots,(L)}(\{\mu_{i,l}\},\{\theta_i\},\{\kappa_l\},\{\delta_{i,l}\})} \sum_{(i,l)\in[K]\times[L]} \Delta_{i,l}q_{i,l},$$

the optimal solution of which is

$$\mathcal{R}^{*,\mathrm{H}}_{(1),\dots,(L)}(\{\mu_{i,l}\},\{\theta_i\},\{\kappa_l\},\{\delta_{i,l}\}) = \Bigg\{\{q_{i,l}\} \in \mathcal{R}^{\mathrm{H}}_{(1),\dots,(L)}(\{\mu_{i,l}\},\{\theta_i\},\{\kappa_l\},\{\delta_{i,l}\}) :$$

$$\sum_{(i,l)\in[K]\times[L]} \Delta_{i,l}q_{i,l} = C^{*,\mathrm{H}}_{(1),\dots,(L)}(\{\mu_{i,l}\},\{\theta_i\},\{\kappa_l\},\{\delta_{i,l}\})\Bigg\}.$$

The necessity of additional terms in PMED-Hinge are discussed in Appendix B. The following theorem, whose proof is in Appendix G, derives a regret upper bound that matches the lower bound in Theorem 2.

**Theorem 6.** (Asymptotic optimality of PMED-Hinge) *Let the solution of the optimal exploration* $\mathcal{R}^{*,\mathrm{H}}_{1,\dots,L}(\{\mu_{i,l}\},\{\theta_i\},\{\kappa_l\},\{\delta_{i,l}\})$ *restricted to* $l \le L$ *is unique at* $(\{\mu^*_{i,l}\},\{\theta^*_i\},\{\kappa^*_l\},\{0\})$. *For any* $\alpha > 0$, $\beta > 0$, *and* $\gamma > 0$, *the regret of PMED-Hinge is bounded as:*

$$\mathbb{E}[\mathrm{Reg}(T)] \le C^*_{1,\dots,L}(\{\mu^*_{i,l}\},\{\theta^*_i\},\{\kappa^*_l\})\log T + o(\log T).$$

Note that, the assumption on the uniqueness of the solution in Theorem 6 is required to achieve an optimal coefficient on the $\log T$ factor. It is not very difficult to derive an $O(\log T)$ regret even though the uniqueness condition is not satisfied. Although our regret bound is not finite-time, the only asymptotic analysis comes from the optimal constant on the top of $\log T$ term (Lemma 11 in Appendix) and it is not very hard to derive an $O(\log T)$ finite-time regret bound.

# 8   Conclusion

By providing a regret lower bound and an algorithm with a matching regret bound, we gave the first complete characterization of a position-based multiple-play multi-armed bandit problem where the quality of the arms and the discount factor of the slots are unknown. We provided a way to compute the optimization problems related to the algorithm, which is of its own interest and is potentially applicable to other bandit problems.

## Acknowledgements

The authors gratefully acknowledge Kohei Komiyama for discussion on a permutation matrix and sincerely thank the anonymous reviewers for their useful comments. This work was supported in part by JSPS KAKENHI Grant Number 17K12736, 16H00881, 15K00031, and Inamori Foundation Research Grant.

## Footnotes

[1]The infimum should take parameters $\theta_i' \kappa_i' \neq \theta_i \kappa_i$ into consideration. However, such parameters can be removed without increasing regret, and thus the infimum over $\theta_i' \kappa_i' = \theta_i \kappa_i$ suffices. This can be understood because the regret bound of the standard $K$-armed bandit problem with expectation of each arm $\mu_i$ is $\sum_{i=2}^K (\log T)/d_{\mathrm{KL}}(\mu_i, \mu_1)$: Arm 1 is drawn without increasing regret, and thus estimation of $\mu_1$ can be arbitrary accurate. In our case placing arms $1, \ldots, L$ into slots $1, \ldots, L$ does not increase the regret, and thus the estimation of the product parameter $\theta_i \kappa_i$ for each $i \in [L]$ is very accurate.

[2]$K \times K$ is not a typo of $K \times L$: $\{q_{i,l}\}$ and $\{\tilde{N}_{i,l}\}$ are sets of $K^2$ variables.

[3]The bi-convex optimization is identical to the one used for obtaining the MLE in PMED.

[4]http://www.gurobi.com

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
