[Supplementary Material]

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

| 2 | 0 | 0 | 1 |
|---|---|---|---|
| 0 | 2 | 0 | 1 |
| 0 | 0 | 2 | 1 |
| 1 | 1 | 1 | 0 |

Figure 3: A $\tilde{N}$ matrix with $K = 4$. Greedily choosing $(1,1), (2,2)$, and $(3,3)$ (of value $2$ each) entries fails to find a maximal matching.

## A  Case in which Greedy Matching Fails

Figure 3 shows a case in which a naive greedy algorithm fails to find a maximal (perfect) matching. Consider the naive greedy algorithm that chooses the largest $\tilde{N}_{i,l}$ among all entries and tries to create a matching iteratively. Such an algorithm tries to focus on the first three diagonal elements of value $2$, after that it stucks, not to yield a perfect matching.

## B  Discussions on Hyperparameters

The hinged version of the algorithm involves three hyperparameters: $\alpha$, $\beta$, and $\gamma$. While $\alpha$ is necessary for assuring the quality of the estimators, we conjecture that the terms $\beta$ and $\gamma$ are just theoretical artifacts and unnecessary: $\beta$ is for stopping the solution $\{q_{i,l}\}$ from diverging (Lemma 22 in the Appendix), which is unlikely to occur for a long time. $\gamma$ is for avoiding a large value of $d_{\mathrm{KL}}(\hat{\mu}_{i,l}(t), \hat{\theta}_i(t)\hat{\kappa}_l(t))$: large deviation principle states that $d_{\mathrm{KL}}(\hat{\mu}_{i,l}(t), \theta_i^*\kappa_l^*) \sim \Theta(1/N_{i,l}(t))$, and thus MLE $(\{\theta_i^*\}, \{\kappa_l^*\})$ is unlikely to behave badly with a moderate value of $N_{i,l}(t)$.

## C  Parameters from KDD Cup 2012 Dataset

Table 1 shows the parameters estimated from the KDD cup 2012 dataset.

## D  Facts

The following facts are frequently used in this paper. Fact 7 is a concentration inequality that bounds the tail probability on the empirical means. Fact 8 is used to bound the KL divergence from below.

**Fact 7.** (The Chernoff bound)
*Let $X_1, \ldots, X_n$ be i.i.d. binary random variables. Let $\hat{X} = \frac{1}{n}\sum_{i=1}^n X_i$ and $\mu = \mathbb{E}[\hat{X}]$. Then, for*

Table 1: Values of $\{\theta_i^*\}$ and $\{\kappa_l^*\}$ estimated from the KDD cup 2012 dataset.

| $K$ | $L$ | $\{\theta_i^*\}$ | $\{\kappa_l^*\}$ |
|---|---|---|---|
| 5 | 3 | $\{0.0463, 0.0135, 0.0127, 0.0106, 0.00629\}$ | $\{1, 0.49, 0.375\}$ |
| 5 | 3 | $\{0.0435, 0.0418, 0.0132, 0.00684, 0.00572\}$ | $\{1, 0.377, 0.187\}$ |
| 6 | 3 | $\{0.0315, 0.0208, 0.0193, 0.0182, 0.0179, 0.0177\}$ | $\{1, 0.534, 0.457\}$ |
| 7 | 3 | $\{0.0405, 0.038, 0.0265, 0.0261, 0.0256, 0.0164, 0.0112\}$ | $\{1, 0.521, 0.46\}$ |
| 10 | 3 | $\{0.0774, 0.0709, 0.0669, 0.0631, 0.043,$ $0.0393, 0.0296, 0.0217, 0.00797, 0.00219\}$ | $\{1, 0.442, 0.285\}$ |
| 5 | 3 | $\{0.0654, 0.0496, 0.0395, 0.0247, 0.0231\}$ | $\{1, 0.359, 0.25\}$ |
| 8 | 3 | $\{0.037, 0.0275, 0.0266, 0.0266, 0.0231, 0.0192, 0.0143, 0.0107\}$ | $\{1, 0.542, 0.393\}$ |
| 5 | 3 | $\{0.147, 0.0343, 0.0272, 0.0222, 0.0166, 0.0162, 0.00966\}$ | $\{1, 0.624, 0.482\}$ |

*any $\epsilon > 0$,*

$$\mathbb{P}(\hat{X} \geq \mu + \epsilon) \leq \exp\left(-d_{\mathrm{KL}}(\mu + \epsilon, \mu)n\right)$$

*and*

$$\mathbb{P}(\hat{X} \leq \mu - \epsilon) \leq \exp\left(-d_{\mathrm{KL}}(\mu - \epsilon, \mu)n\right).$$

**Fact 8.** (The Pinsker's inequality)
*For $p, q \in (0, 1)$, the KL divergence between two Bernoulli distributions is bounded as:*

$$d_{\mathrm{KL}}(p, q) \geq 2(p - q)^2.$$

## E   Proof of Theorem 4

*Proof of Theorem 4.* Consider the bipartite graph that we described in Section 4.2. By assumption, the sums of every row and column are identical, and let $r_{\mathrm{sum}}$ be that value. Let $I$ and $L$ be the left and right nodes, respectively. From Hall's marriage theorem [15], a bipartite graph has a perfect matching iff

$$|I^{\mathrm{sub}}| \leq |N_G(I^{\mathrm{sub}})| \tag{3}$$

for every subset $I^{\mathrm{sub}}$ of $I$, where $N_G(x)$ is the neighbors of $x$. We prove inequality (3) by contradiction. Assume that there exist an $I^{\mathrm{sub}}$ such that $|I^{\mathrm{sub}}| > |N_G(I^{\mathrm{sub}})|$. Then,

$$\frac{1}{N_G(I^{\mathrm{sub}})} \sum_{l \in N_G(I^{\mathrm{sub}})} \sum_{i \in [K]} r_{i,l} \geq \frac{1}{N_G(I^{\mathrm{sub}})} \sum_{i \in I^{\mathrm{sub}}} \sum_{l \in [K]} r_{i,l} = \frac{|I^{\mathrm{sub}}|}{|N_G(I^{\mathrm{sub}})|} r_{\mathrm{sum}} > r_{\mathrm{sum}}.$$

and thus at least one of the columns has a sum larger than $r_{\mathrm{sum}}$, which contradicts the fact that the sum of every row and column is $r_{\mathrm{sum}}$. □

## F   Proofs of Regret Lower Bound

In this section, we prove Lemma 1 and Theorem 2. In the following, we frequently denote $\{\mathcal{A}, \mathcal{B}\}$ instead of $\{\mathcal{A} \cap \mathcal{B}\}$ for two events $\mathcal{A}$ and $\mathcal{B}$.

*Proof of Lemma 1.* The technique here is inspired from Theorem 1 in Lai and Robbins [28]. Let $\{\theta_i'\}, \{\kappa_l'\}$ be another set of parameters such that $\theta_i^* \kappa_i^* = \theta_i' \kappa_i'$ and there exists $i \neq j, i \in [L], j \in [K]$ such that $(\theta_i^* - \theta_j^*)(\theta_i' - \theta_j') < 0$. Let $\mu_{i,l}^* = \theta_i^* \kappa_l^*$ and $\mu_{i,l}' = \theta_i' \kappa_l'$. Let $(i)' \in [K]$ be the index of $i$-th best parameters among $\{\theta_i'\}$. With these parameters, there exists $i \in [L]$ such that $(i)' \neq i$ (i.e., the list of top-$L$ arms is different from the ones of the true parameters). We consider a modified bandit problem with this parameters.

Let $x_{i,l}^m \in \{0, 1\}$ is the reward of the $m$-th observation of arm $i$ in slot $l$. Let

$$\widehat{\mathrm{KL}}_{i,l}(n) = \sum_{m=1}^{n} \log\left(\frac{x_{i,l}^m \mu_{i,l}^* + (1 - x_{i,l}^m)(1 - \mu_{i,l}^*)}{x_{i,l}^m \mu_{i,l}' + (1 - x_{i,l}^m)(1 - \mu_{i,l}')}\right),$$

and $\widehat{\mathrm{KL}} = \sum_{(i,l) \in [K] \times [L]} \widehat{\mathrm{KL}}_{i,l}(N_{i,l}(t))$. Let $\mathbb{P}'$ and $\mathbb{E}'$ be the probability and the expectation with respect to the modified game, respectively. Then, for any event $\mathcal{E}$,

$$\mathbb{P}'[\mathcal{E}] = \mathbb{E}\left[\mathbf{1}[\mathcal{E}] \exp\left(-\widehat{\mathrm{KL}}\right)\right] \tag{4}$$

holds. Now, let us define the following events:

$$\mathcal{D}_1 = \left\{ \sum_{(i,l) \in [K] \times [L]} N_{i,l}(t) d_{\mathrm{KL}}(\mu_{i,l}^*, \mu_{i,l}') < (1 - \epsilon) \log T, \bigcap_{i \in [L] : i \neq (i)'} N_{(i)',i}(t) < \sqrt{T} \right\},$$

$$\mathcal{D}_2 = \left\{ \widehat{\mathrm{KL}} \leq \left(1 - \frac{\epsilon}{2}\right) \log T \right\},$$

$$\mathcal{D}_{12} = \mathcal{D}_1 \cap \mathcal{D}_2,$$

$$\mathcal{D}_{1\backslash 2} = \mathcal{D}_1 \cap \mathcal{D}_2^c.$$

Firstly, we show $\mathbb{P}[\mathcal{D}_{12}] = o(1)$. From (4),

$$\mathbb{P}'[\mathcal{D}_{12}] \geq \mathbb{E}\left[\mathbf{1}[\mathcal{D}_{12}] \exp\left(-\left(1 - \frac{\epsilon}{2}\right)\log T\right)\right] = T^{-(1-\epsilon/2)}\mathbb{P}[\mathcal{D}_{12}].$$

By using this we have

$$\mathbb{P}[\mathcal{D}_{12}] \leq T^{(1-\epsilon/2)}\mathbb{P}'[\mathcal{D}_{12}]$$

$$\leq T^{(1-\epsilon/2)}\mathbb{P}'\left[\bigcap_{i\in[L]:i\neq(i)'}\{N_{(i)',i}(t) < \sqrt{T}\}\right]$$

$$\leq T^{(1-\epsilon/2)}\sum_{i\in[L]:i\neq(i)'}\mathbb{P}'\left[T - N_{(i)',i}(t) > T - \sqrt{T}\right]$$

$$\leq T^{(1-\epsilon/2)}\sum_{i\in[L]:i\neq(i)'}\frac{\mathbb{E}'[T - N_{(i)',i}(t)]}{T - \sqrt{T}} \qquad \text{(by the Markov inequality).} \qquad (5)$$

Since this algorithm is strongly consistent, $\mathbb{E}'[T - N_{(i)',i}(t)] \to o(T^a)$ for any $a > 0$. Therefore, the RHS of the last line of (5) is $o(T^{a-\epsilon/2})$, which, by choosing sufficiently small $a$, converges to zero as $T \to \infty$. In summary, $\mathbb{P}[\mathcal{D}_{12}] = o(1)$.

Secondly, we show $\mathbb{P}[\mathcal{D}_{1\backslash 2}] = o(1)$. We have

$$\mathbb{P}[\mathcal{D}_{1\backslash 2}] = \mathbb{P}\left[\sum_{(i,l)\in[K]\times[L]} N_{i,l}(t)d_{\mathrm{KL}}(\mu_{i,l}, \mu'_{i,l}) < (1 - \epsilon)\log T,\right.$$

$$\left.\bigcap_{i\in[L]:i\neq(i)'}N_{(i)',i}(t) < \sqrt{T}, \quad \sum_{(i,l)\in[K]\times[L]}\widehat{\mathrm{KL}}_i(N_{i,l}(t)) > \left(1 - \frac{\epsilon}{2}\right)\log T\right].$$

Note that

$$\max_{1\leq n\leq N}\widehat{\mathrm{KL}}_{i,l}(n)$$

is the maximum of the sum of positive-mean random variables, and thus converges to is average (c.f., Bubeck [5, Lemma 10.5]). Namely,

$$\lim_{N\to\infty}\max_{1\leq n\leq N}\frac{\widehat{\mathrm{KL}}_{i,l}(n)}{N} = d_{\mathrm{KL}}(\mu^*_{i,l}, \mu'_{i,l})$$

almost surely. Therefore,

$$\lim_{T\to\infty}\frac{\max_{\{N_{i,l}(t)\}\in\mathbb{N}^N, \sum_{(i,l)\in[K]\times[L]}N_{i,l}(t)d_{\mathrm{KL}}(\mu^*_{i,l},\mu'_{i,l})<(1-\epsilon)\log T}\sum_{(i,l)\in[K]\times[L]}\widehat{\mathrm{KL}}_i(N_{i,l}(t))}{\log T} = 1-\epsilon$$

almost surely. By using this fact and $1 - \epsilon/2 > 1 - \epsilon$, we have

$$\mathbb{P}\left[\max_{\{N_{i,l}(t)\}\in\mathbb{N}^{KL}, \sum_{(i,l)\in[K]\times[L]}N_{i,l}(t)d_{\mathrm{KL}}(\mu^*_{i,l},\mu'_{i,l})<(1-\epsilon)\log T}\sum_{(i,l)\in[K]\times[L]}\widehat{\mathrm{KL}}_i(N_{i,l}(t)) > \left(1 - \frac{\epsilon}{2}\right)\log T\right] = o(1).$$

In summary, we obtain $\mathbb{P}\left[\mathcal{D}_{1\backslash 2}\right] = o(1)$.

We here have

$$\mathcal{D}_1 = \left\{\sum_{(i,l)\in[K]\times[L]}N_{i,l}(t)d_{\mathrm{KL}}(\mu^*_{i,l}, \mu'_{i,l}) < (1 - \epsilon)\log T\right\} \cap \left\{\bigcap_{i\in[L]:i\neq(i)'}N_{(i)',i}(t) < \sqrt{T}\right\}$$

$$\supseteq \left\{\sum_{(i,l)\in[K]\times[L]}N_{i,l}(t)d_{\mathrm{KL}}(\mu_{i,l}, \mu'_{i,l}) + \sum_{i\in[L]:i\neq(i)'}\frac{(1-\epsilon)\log T}{\sqrt{T}}N_{(i)',i}(t) < (1 - \epsilon)\log T\right\},$$

where we used the fact that $\{A < C\} \cap \{B < C\} \supseteq \{A + B < C\}$ for $A, B > 0$ in the last line. Note that, by using the result of the previous steps, $\mathbb{P}[\mathcal{D}_1] = \mathbb{P}[\mathcal{D}_{12}] + \mathbb{P}[\mathcal{D}_{1\backslash 2}] = o(1)$. By using the complementary of this fact,

$$
\mathbb{P}\left[ \sum_{(i,l)\in[K]\times[L]} N_{i,l}(t) d_{\mathrm{KL}}(\mu_{i,l}^*, \mu_{i,l}') + \sum_{i\in[L]:i\neq(i)'} \frac{(1-\epsilon)\log T}{\sqrt{T}} N_{(i)',i}(t) \geq (1-\epsilon)\log T \right]
$$
$$
\geq \mathbb{P}[\mathcal{D}_1^c] = 1 - o(1).
$$

Using the Markov inequality yields

$$
\mathbb{E}\left[ \sum_{(i,l)\in[K]\times[L]} N_{i,l}(t) d_{\mathrm{KL}}(\mu_{i,l}^*, \mu_{i,l}') + \frac{(1-\epsilon)\log T}{\sqrt{T}} \sum_{i\in[L]:i\neq(i)'} N_{(i)',i}(t) \right] \geq (1-\epsilon)(1-o(1))\log T.
$$
(6)

Because $\mathbb{E}[N_{(i)',i}(t)]$ is subpolynomial as a function of $T$ due to the consistency, the second term in LHS of (6) is $o(1)$ and thus negligible. Lemma 1 follows from the fact that (6) holds for sufficiently small $\epsilon$ and arbitrary $\{\theta_i', \kappa_l'\}$. $\qquad\square$

*Proof of Theorem 2.* Assume that there exists $\delta > 0$ and a sequence $T_1 < T_2 < T_3 < \cdots$ such that for all $s$

$$
\mathbb{E}[\mathrm{Reg}(T_s)] < (1-\delta) C_{1,\dots,L}^*(\{\mu_{i,l}^*\}, \{\theta_i^*\}, \{\kappa_l^*\}) \log T_s ,
$$

that is,

$$
\sum_{(i,l)\in[K]\times[L]} \Delta_{i,l} \frac{\mathbb{E}[N_{i,l}(T_s)]}{(1-\delta)\log T_s} < C_{1,\dots,L}^*(\{\mu_{i,l}^*\}, \{\theta_i^*\}, \{\kappa_l^*\}) .
$$

From the definition of $C_{1,\dots,L}^*$, there exists $\{\theta_i^s\} \in \mathcal{T}_{1,\dots,L}^c$ such that

$$
\sum_{(i,l)\in[K]\times[L]} d_{\mathrm{KL}}\left( \theta_i^* \kappa_l^*, \theta_i^s \frac{\theta_l^* \kappa_l^*}{\theta_l^s} \right) \frac{\mathbb{E}[N_{i,l}(T_s)]}{(1-\delta)\log T_s} < 1
$$

Since $\mathcal{T}_{1,\dots,L}^c$ is compact, there exists a subsequence $s_0 < s_1 < \cdots$ such that $\lim_{u\to\infty}\{\theta_i^{s_u}\} = \{\theta_i\}$ for some $\{\theta_i\} \in \mathcal{T}_{1,\dots,L}^c$. Therefore from the lower semicontinuity of the divergence we obtain

$$
1 \geq \sum_{(i,l)\in[K]\times[L]} \liminf_{u\to\infty} \frac{\mathbb{E}[N_i(T_{s_u})]}{(1-\delta)\log T_{s_u}} d_{\mathrm{KL}}\left( \theta_i^* \kappa_l^*, \theta_i^{s_u} \frac{\theta_l^* \kappa_l^*}{\theta_l^{s_u}} \right)
$$
$$
\geq \sum_{(i,l)\in[K]\times[L]} \liminf_{s\to\infty} \frac{\mathbb{E}[N_i(T_s)]}{(1-\delta)\log T_s} d_{\mathrm{KL}}\left( \theta_i^* \kappa_l^*, \theta_i \frac{\theta_l^* \kappa_l^*}{\theta_l} \right) ,
$$

which contradicts Lemma 1.

$\qquad\square$

## G Main Regret Bound: Proof of Theorem 6

In this section, we prove the asymptotic optimality of PMED-Hinge. First, we define the following events that are important in bounding regret. Let $\hat{i}(t)$ be the $i$-th largest arm based on $\hat{\theta}_i(t)$ (ties are broken arbitrarily).

$$
\mathcal{W}(t) = \bigcup_{i\in[L]} \left\{ \hat{i}(t) \neq i \right\}
$$

Henceforth, we abbreviate the event that some $L$-allocation that includes $(i, l) \in [K] \times [L]$ is put into $L_N$ to "pair $(i, l)$ is put into $L_N$". Let $\mathcal{J}_{i,l}(t)$ be the event that pair $(i, l)$ is put into $L_N$ at round

$t$. Let $\mathcal{X}(t)$ is the event that at least one arm is put into $L_N$ before Line 18 in Algorithm 1. Namely,

$$\mathcal{X}(t) = \Bigg\{ \bigcup_{(i,l)\in[K]\times[L]} \{N_{i,l}(t) < \alpha\sqrt{\log t}\} \cup \bigcup_{(i\neq j)\in[K]\times[K]} \{|\hat{\theta}_i(t) - \hat{\theta}_j(t)| < \beta/(\log\log t)\}$$

$$\cup \bigcup_{(l\neq m)\in[K]\times[K]} \{|\hat{\kappa}_l(t) - \hat{\kappa}_m(t)| < \beta/(\log\log t)\} \cup \bigcup_{(i,l)\in[K]\times[L]} \Big\{ d_{\mathrm{KL}}(\hat{\mu}_{i,l}(t), \hat{\theta}_i(t)\hat{\kappa}_l(t)) \geq f(N_{i,l}(t)) \Big\} \Bigg\}.$$

Let $\mathcal{Y}_{i,l}(t)$ be the event that pair $(i,l)$ is put into $L_N$ in Line 18 of Algorithm 1. Moreover, let

$$\mathcal{Z}_\delta(t) = \bigcap_{i\in[K]} \{|\hat{\theta}_i(t) - \theta_i^*| < \delta\} \cap \bigcap_{l\in[L]} \{|\hat{\kappa}_l(t) - \kappa_l^*| < \delta\}.$$

That is, the estimator $\{\hat{\theta}_i(t), \hat{\kappa}_l(t)\}$ is sufficiently close to the set of true values.

By using $\Delta_{i,l} \leq 1$, the regret is decomposed into the following terms:

$$\mathrm{Reg}(T) = \sum_{(i,l)\in[K]\times[L]} \Delta_{i,l} \sum_{t=1}^{T} \mathbf{1}[I_i(t) = l]$$

$$\leq \sum_{(i,l)\in[K]\times[L]} \Delta_{i,l} \sum_{t=1}^{T} \mathbf{1}[\mathcal{J}_{i,l}(t)] + K^2$$

$$\leq K \sum_{t=1}^{T} \mathbf{1}[\mathcal{X}(t), \mathcal{J}_{i,l}(t)] + \Delta_{i,l} \sum_{t=1}^{T} \mathbf{1}[\mathcal{X}^c(t), \mathcal{Y}_{i,l}(t)] + K^2$$

$$\leq K \sum_{t=1}^{T} \mathbf{1}[\mathcal{X}(t), \mathcal{J}_{i,l}(t)] + K \sum_{t=1}^{T} \mathbf{1}[\mathcal{X}^c(t), \mathcal{W}(t)] + \sum_{(i,l)\in[K]\times[L]} \Delta_{i,l} \sum_{t=1}^{T} \mathbf{1}[\mathcal{W}^c(t), \mathcal{Y}_{i,l}(t), \mathcal{Z}_\delta(t)]$$

$$+ \sum_{(i,l)\in[K]\times[L]} \sum_{t=1}^{T} \mathbf{1}[\mathcal{X}^c(t), \mathcal{W}^c(t), \mathcal{Y}_{i,l}(t), \mathcal{Z}_\delta^c(t)] + K^2 \tag{7}$$

The following lemmas bound each term of (7), and combining them completes the proof.

**Lemma 9.** (Sublog exploration) *The following inequality holds:*

$$\sum_{t=1}^{T} \sum_{(i,l)\in[K]\times[L]} \mathbb{P}\left[\mathcal{X}(t), \mathcal{J}_{i,l}(t)\right] = o(\log T)$$

**Lemma 10.** (Misidentification) *The following inequality holds:*

$$\sum_{t=1}^{T} \mathbb{P}\left[\mathcal{X}^c(t), \mathcal{W}(t)\right] = O(1)$$

**Lemma 11.** (Leading term) *There exists a continuous function $\epsilon(\delta)$ such that $\epsilon(\delta) \to 0$ as $\delta \to 0$, and the following inequality holds:*

$$\sum_{t=1}^{T} \mathbf{1}\left[\mathcal{W}^c(t), \mathcal{Y}_{i,l}(t), \mathcal{Z}_\delta(t)\right] = (1 + \epsilon(\delta))R_{i,l}^* \log T + 1.$$

*where $R_{i,l}^*$ is the $(i,l)$ entry of the optimal solution $\mathcal{R}_{1,\dots,L}^*(()\{\mu_{i,l}^*\}, \{\theta_i^*\}, \{\kappa_i^*\})$.*

**Lemma 12.** (Case in which estimation quality is low) *The following inequality holds:*

$$\sum_{t=1}^{T} \mathbb{P}\left[\mathcal{X}^c(t), \mathcal{W}^c(t), \mathcal{Y}_{i,l}(t), \mathcal{Z}_\delta^c(t)\right] = o(\log T).$$

The following sections prove Lemmas 9–12. We use $\hat{\mu}_{i,l}^n$ to denote $\hat{\mu}_{i,l}(t)$ when $N_{i,l}(t) = n$.

# H  Proof of Lemma 9

**Lemma 13.** (Convergence of the hinge of MLE) *Let*

$$\mathcal{C}(t) = \bigcap_{(i,l)\in[K]\times[L]} \left\{ d_{\mathrm{KL}}(\hat{\mu}_{i,l}(t), \hat{\theta}_i(t)\hat{\kappa}_l(t)) \le f(N_{i,l}(t)) \right\}. \tag{8}$$

*The following inequality holds of any $(i,l) \in [K] \times [L]$:*

$$\sum_{t=1}^{T} \mathbb{P}\left[ \mathcal{J}_{i,l}(t), \mathcal{C}^c(t) \right] = O(1).$$

*Proof of Lemma 13.* Let $N^C(t)$ be the number of rounds before $t$ such that pair $(i,l)$ is put into $L_N$ by Line 10 of Algorithm 1. Note that $\{\mathcal{J}_{i,l}(t), \mathcal{C}^c(t), N^C(t) = n\}$ occurs at most twice because if $\mathcal{C}^c(t)$ occurs then $(i,l)$ is put into $L_N$ by Line 10. By using this fact, we obtain

$$\sum_{t=1}^{T} \mathbb{P}\left[ \mathcal{J}_{i,l}(t), \mathcal{C}^c(t) \right]$$

$$\le 2\sum_{n=1}^{T} \mathbb{P}\left[ \bigcup_{t=n}^{T} \{\mathcal{J}_{i,l}(t), \mathcal{C}^c(t), N^C(t) = n\} \right]$$

$$\le 2\sum_{n=1}^{T} \mathbb{P}\left[ \bigcup_{t=n}^{T} \{\mathcal{J}_{i,l}(t), \mathcal{C}^c(t), N_{i,l}(t) \ge n\} \right]$$

$$\le 2\sum_{n=1}^{T} \sum_{(i,l)\in[K]\times[L]} \sum_{n'=n}^{T} \mathbb{P}\left[ d_{\mathrm{KL}}(\hat{\mu}_{i,l}^{n'}, \theta_i^* \kappa_l^*) > f(n') \right] \quad \le 2\sum_{n=1}^{T} \sum_{(i,l)\in[K]\times[L]} \sum_{n'=n}^{T} e^{-n'f(n')} = O(1)$$

$\square$

where we have used the facts that $N^C(t) = n \Rightarrow \bigcap_{(i,l)} \{N_{i,l}(t) \ge n\}$ and $\mathcal{C}^c(t)$ implies $\bigcup_{(i,l)} \{d_{\mathrm{KL}}(\hat{\mu}_{i,l}^{n'}, \theta_i^* \kappa_l^*) > f(n')\}$.

**Lemma 14.** *For any $i \ne j \in [K]$, the following inequality holds:*

$$\sum_{t=1}^{T} \mathbb{P}\left[ \bigcap_{(i',l')\in[K]\times[L]} N_{i',l'}(t) \ge \alpha\sqrt{\log t}, \mathcal{J}_{i,l}(t), \mathcal{C}(t), |\hat{\theta}_i(t) - \hat{\theta}_j(t)| < \beta/(\log\log t) \right] = O(1).$$

*Proof of Lemma 14.* Let $\Delta_\theta = |\theta_i^* - \theta_j^*| > 0$. Note that

$$\left\{ |\hat{\theta}_i(t) - \hat{\theta}_j(t)| < \beta/(\log\log t), t \ge e^{e^{5\beta/\Delta_\theta}} \right\}$$

implies $|\hat{\theta}_i(t) - \hat{\theta}_j(t)| \le \Delta_\theta/5$. Moreover,

$$\left\{ N_{i',l'}(t) > \alpha\sqrt{\log t}, t > e^{\left(\frac{(5^2\gamma)^2}{4\alpha\Delta_\theta^2}\right)^2} \right\}$$

implies that $2(\Delta_\theta/5)^2 > f(N_{i',l'}(t))$. Let $N^D(t)$ be the number of rounds before $t$ such that the arms are put into $L_N$ by Line 9. Then $\{\mathcal{J}_{i,l}(t), N^D(t) = n\}$ occurs at most twice. By using these,

we obtain

$$\sum_{t=1}^{T} \mathbb{P}\left[\bigcap_{(i',l')\in[K]\times[L]} N_{i',l'}(t) > \alpha\sqrt{\log t}, |\hat{\theta}_i(t) - \hat{\theta}_j(t)| < \beta/(\log\log t), \mathcal{J}_{i,l}(t), \mathcal{C}(t)\right]$$

$$\leq \sum_{n=1}^{T} \mathbb{P}\left[\bigcup_{t=1}^{T}\left\{\bigcap_{(i',l')\in[K]\times[L]} N_{i',l'}(t) > \alpha\sqrt{\log t}, |\hat{\theta}_i(t) - \hat{\theta}_j(t)| < \beta/(\log\log t), N^D(t) = n\right\}, \mathcal{J}_{i,l}(t), \mathcal{C}(t)\right]$$

$$\leq \max\left(e^{\left(\frac{(5^2\gamma)^2}{4\alpha\Delta_\theta^2}\right)^2}, e^{e^{5\beta/\Delta_\theta}}\right) + \sum_{n=1}^{T}\mathbb{P}\left[\bigcup_{t=1}^{T}\left\{\mathcal{C}(t), |\hat{\theta}_i(t) - \hat{\theta}_j(t)| < \Delta_\theta/5, 2(\Delta_\theta/5)^2 > f(N_{i,l}(t)), N^D(t) = n\right\}\right]$$

$$\leq O(1) + \sum_{n=1}^{T}\mathbb{P}\left[\bigcup_{t=1}^{T}\left\{d_{\mathrm{KL}}(\hat{\mu}_{i,1}(t), \hat{\theta}_i(t)) < 2(\Delta_\theta/5)^2, d_{\mathrm{KL}}(\hat{\mu}_{j,1}(t), \hat{\theta}_j(t)) < 2(\Delta_\theta/5)^2, \right.\right.$$

$$\left.\left. |\hat{\theta}_i(t) - \hat{\theta}_j(t)| < \Delta_\theta/5, N^D(t) = n\right\}\right]$$

$$\leq O(1) + \sum_{n=1}^{T}\mathbb{P}\left[\bigcup_{t=1}^{T}\left\{(\hat{\mu}_{i,1}(t) - \hat{\theta}_i(t))^2 < (\Delta_\theta/5)^2, (\hat{\mu}_{j,1}(t) - \hat{\theta}_j(t))^2 < (\Delta_\theta/5)^2, \right.\right.$$

$$\left.\left. |\hat{\theta}_i(t) - \hat{\theta}_j(t)| < \Delta_\theta/5, N^D(t) = n\right\}\right]$$

<p align="center">(by Pinsker's inequality)</p>

$$\leq O(1) + \sum_{n=1}^{T}\mathbb{P}\left[\bigcup_{t=1}^{T}\left\{|\hat{\mu}_{i,1}(t) - \hat{\theta}_i(t)| < \Delta_\theta/5, |\hat{\mu}_{j,1}(t) - \hat{\theta}_j(t)| < \Delta_\theta/5, |\hat{\theta}_i(t) - \hat{\theta}_j(t)| < \Delta_\theta/5, N^D(t) = n\right\}\right]$$

$$\leq O(1) + \sum_{n=1}^{T}\mathbb{P}\left[\bigcup_{t=1}^{T}\left\{|\hat{\mu}_{i,1}(t) - \theta_i^*| > \Delta_\theta/5 \cup |\hat{\mu}_{j,1}(t) - \theta_j^*| > \Delta_\theta/5\right\}\right]$$

<p>(by the triangular inequality and the definition of $\Delta_\theta$.)</p>

$$\leq O(1) + \sum_{n=1}^{T}\mathbb{P}\left[\bigcup_{t=1}^{T}\left\{|\hat{\mu}_{i,1}(t) - \theta_i^*| > \Delta_\theta/5\right\}\right] + \sum_{n=1}^{T}\mathbb{P}\left[\bigcup_{t=1}^{T}\left\{|\hat{\mu}_{j,1}(t) - \theta_j^*| > \Delta_\theta/5\right\}\right]. \quad (9)$$

Here,

$$\sum_{n=1}^{T}\mathbb{P}\left[\bigcup_{t=1}^{T}\left\{|\hat{\mu}_{i,1}(t) - \theta_i^*| > \Delta_\theta/5, N^D(t) = n\right\}\right]$$

$$\leq \sum_{n=1}^{T}\sum_{n'=n}^{T}\mathbb{P}\left[|\hat{\mu}_{i,1}^{n'} - \theta_i^*| > \Delta_\theta/5\right]$$

<p align="center">(by the fact that $N^D(t) = n$ implies $N_{i,l}(t) \geq n$.)</p>

$$\leq \sum_{n=1}^{T}\sum_{n'=n}^{T} e^{2n'(\Delta_\theta/5)^2} = O(1),$$

<p align="center">(by Chernoff bound and Pinsker's inequality)</p>

<p align="right">(10)</p>

and the same inequality also holds for $j$, which completes the proof. $\qquad\square$

**Lemma 15.** *For any $l \neq m \in [K]$, the following inequality holds:*

$$\sum_{t=1}^{T}\mathbb{P}\left[\bigcap_{(i',l')\in[K]\times[L]} N_{i',l'}(t) \geq \alpha\sqrt{\log t}, |\hat{\kappa}_l(t) - \hat{\kappa}_m(t)| < \beta/(\log\log t), \mathcal{J}_{i,l}(t), \mathcal{C}(t)\right] = O(1).$$

*Proof of Lemma 15.* Let $\Delta_\kappa = \min_{l \neq m} \theta_1 |\kappa_l^* - \kappa_m^*|$. Similar to Lemma 14, we have

$$\sum_{t=1}^{T} \mathbb{P}\left[\bigcap_{(i',l') \in [K] \times [L]} N_{i',l'}(t) \geq \alpha\sqrt{\log t}, |\hat{\kappa}_l(t) - \hat{\kappa}_m(t)| < \beta/(\log\log t), \mathcal{J}_{i,l}(t), \mathcal{C}(t)\right]$$

$$\leq \sum_{n=1}^{T} \mathbb{P}\left[\bigcup_{t=1}^{T}\left\{\bigcap_{(i',l') \in [K] \times [L]} N_{i',l'}(t) > \alpha\sqrt{\log t}, |\hat{\kappa}_l(t) - \hat{\kappa}_m(t)| < \beta/(\log\log t), N^D(t) = n\right\}, \mathcal{J}_{i,l}(t), \mathcal{C}(t)\right]$$

$$\leq \max\left(e^{\left(\frac{(5^2\gamma)^2}{4\alpha\Delta_\kappa^2}\right)^2}, e^{e^{5\beta/\Delta_\kappa}}\right) + \sum_{n=1}^{T} \mathbb{P}\left[\bigcup_{t=1}^{T}\left\{\mathcal{C}(t), |\hat{\kappa}_l(t) - \hat{\kappa}_m(t)| < \Delta_\kappa/5, 2(\Delta_\kappa/5)^2 > f(N_{i,l}(t)), N^D(t) = n\right\}\right]$$

$$\leq O(1) + \sum_{n=1}^{T} \mathbb{P}\left[\bigcup_{t=1}^{T}\left\{d_{\mathrm{KL}}(\hat{\mu}_{1,l}(t), \hat{\theta}_1(t)\hat{\kappa}_l(t)) < 2(\Delta_\kappa/5)^2, d_{\mathrm{KL}}(\hat{\mu}_{1,m}(t), \hat{\theta}_1(t)\hat{\kappa}_m(t)) < 2(\Delta_\kappa/5)^2, \right.\right.$$

$$\left.\left.|\hat{\kappa}_l(t) - \hat{\kappa}_m(t)| < \Delta_\kappa/5, N^D(t) = n\right\}\right]$$

$$\leq O(1) + \sum_{n=1}^{T} \mathbb{P}\left[\bigcup_{t=1}^{T}\left\{(\hat{\mu}_{1,l}(t) - \hat{\theta}_1(t)\hat{\kappa}_l(t))^2 < (\Delta_\kappa/5)^2, (\hat{\mu}_{1,m}(t) - \hat{\theta}_1(t)\hat{\kappa}_m(t))^2 < (\Delta_\kappa/5)^2, \right.\right.$$

$$\left.\left.|\hat{\kappa}_l(t) - \hat{\kappa}_m(t)| < \Delta_\kappa/5, N^D(t) = n\right\}\right]$$

$$\leq O(1) + \sum_{n=1}^{T} \mathbb{P}\left[\bigcup_{t=1}^{T}\left\{|\hat{\mu}_{1,l}(t) - \hat{\theta}_1(t)\hat{\kappa}_l(t)| < \Delta_\kappa/5, \right.\right.$$

$$\left.\left.|\hat{\mu}_{1,m}(t) - \hat{\theta}_1(t)\hat{\kappa}_m(t)| < \Delta_\kappa/5, \theta_1|\hat{\kappa}_l(t) - \hat{\kappa}_m(t)| < \Delta_\kappa/5, N^D(t) = n\right\}\right]$$

$$\leq O(1) + \sum_{n=1}^{T} \mathbb{P}\left[\bigcup_{t=1}^{T}\left\{|\hat{\mu}_{1,l}(t) - \mu_{1,l}^*| > \Delta_\kappa/5 \cup |\hat{\mu}_{1,m}(t) - \mu_{1,m}^*| > \Delta_\kappa/5\right\}\right]$$

$$\leq O(1) + \sum_{n=1}^{T} \mathbb{P}\left[\bigcup_{t=1}^{T}\left\{|\hat{\mu}_{1,l}(t) - \mu_{1,l}^*| > \Delta_\kappa/5\right\}\right] + \sum_{n=1}^{T} \mathbb{P}\left[\bigcup_{t=1}^{T}\left\{|\hat{\mu}_{1,m}(t) - \mu_{1,m}^*| > \Delta_\kappa/5\right\}\right].$$

$$\tag{11}$$

Here,

$$\sum_{n=1}^{T} \mathbb{P}\left[\bigcup_{t=1}^{T}\left\{|\hat{\mu}_{1,l}(t) - \mu_{1,l}^*| > \Delta_\kappa/5, N^D(t) = n\right\}\right]$$

$$\leq \sum_{n=1}^{T}\sum_{n'=n}^{T} \mathbb{P}\left[|\hat{\mu}_{1,l}^{n'} - \mu_{1,l}^*| > \Delta_\kappa/5\right]$$

$$\leq \sum_{n=1}^{T}\sum_{n'=n}^{T} e^{2n'(\Delta_\kappa/5)^2} = O(1),$$

$$\tag{12}$$

and the same inequality also holds for $m$, which completes the proof. $\qquad\square$

*Proof of 9.* The event

$$\{N_{i',l'}(t) < \alpha\sqrt{\log t}, \mathcal{J}_{i',l'}(t), N_{i',l'}(t) = n\}$$

occurs at most twice because if it occurs then pair $(i', l')$ is put into $L_N$ immediately. Taking these into consideration,

$$\sum_{t=1}^{T} \mathbb{P}\left[\mathcal{X}(t), \mathcal{J}_{i,l}(t)\right]$$

$$\leq 2\sum_{n=1}^{T} \mathbb{P}\left[\bigcup_{t=n}^{T}\left\{\bigcup_{(i',l')\in[K]\times[L]} N_{i',l'}(t) < \alpha\sqrt{\log T}, N_{i',l'}(t) = n\right\}\right] + \sum_{t=1}^{T}\mathbb{P}\left[\mathcal{J}_{i,l}(t), \mathcal{C}^c(t)\right]$$

$$+\sum_{t=1}^{T}\sum_{i\neq j}\mathbb{P}\left[\bigcap_{(i',l')\in[K]\times[L]} N_{i',l'}(t) \geq \alpha\sqrt{\log t}, |\hat{\theta}_i(t) - \hat{\theta}_j(t)| < \beta/(\log\log t), \mathcal{J}_{i,l}(t), \mathcal{C}(t)\right]$$

$$+\sum_{t=1}^{T}\sum_{l\neq m}\mathbb{P}\left[\bigcap_{(i',l')\in[K]\times[L]} N_{i',l'}(t) \geq \alpha\sqrt{\log t}, |\hat{\kappa}_l(t) - \hat{\kappa}_m(t)| < \beta/(\log\log t), \mathcal{J}_{i,l}(t), \mathcal{C}(t)\right]$$

$$(13)$$

The first term of (13) is $O(\sqrt{\log T})$, and the second, third, and fourth terms of (13) are $O(1)$ by Lemma 13, 14, and 15 respectively. $\qquad\square$

## I  Proof of Lemma 10

Remember that $\hat{1}(t), \ldots, \hat{L}(t)$ are the empirical top-$L$ arm based on MLE. The following lemma guarantees the convergence of $\hat{\mu}_{\hat{l}(t),l}(t)$ for each $l \in [L]$.

**Lemma 16.** (Convergence of the estimated best arm) *Let*

$$\mathcal{P}(t) = \bigcup_{l\in[L]}\left\{d_{\mathrm{KL}}(\hat{\mu}_{\hat{l}(t),l}(t), \theta^*_{\hat{l}(t)}\kappa^*_l)\} > f(N_{\hat{l}(t),l}(t))\right\}.$$

*Then, the following inequality holds:*

$$\sum_{t=1}^{T}\mathbb{P}\left[\mathcal{P}(t)\right] = O(1). \tag{14}$$

*Proof of Lemma 16.* Let $(1), \ldots, (L)$ be $L$ distinct elements among $[K]$. Note that,

$$\left\{\hat{1}(t) = (1), \ldots, \hat{L}(t) = (L), \bigcup_{l\in[L]} d_{\mathrm{KL}}(\hat{\mu}_{(l),l}(t), \theta^*_{(l)}\kappa^*_l)\} > f(N_{(l),l}(t)), \bigcup_{l\in[L]} N_{(l),l}(t) \leq n\right\}$$

occurs at most $K^2 n$ rounds because $((1),1),\ldots,((L),L)$ are put into $L_N$ by Line 22 of Algorithm 1 under this event and $|L_N|$ is at most $K^2$. By using this, we have

$$\mathbb{E}\left[\hat{1}(t) = (1),\ldots,\hat{L}(t) = (L), \bigcup_{l\in[L]} d_{\mathrm{KL}}(\hat{\theta}_{(l)}(t)\hat{\kappa}_l(t), \theta^*_{(l)}\kappa^*_l)\} > f(N_{(l),l}(t))\right]$$

$$\leq \sum_{n=2}^{\infty} K^2 n \mathbb{P}\left[\bigcup_{t=1}^{T}\left(\bigcup_{l\in[L]} d_{\mathrm{KL}}(\hat{\mu}_{(l),l}(t), \theta^*_{(l)}\kappa^*_l)\} > f(N_{(l),l}(t)), \bigcup_{l\in[L]}\{N_{(l),l}(t) \leq n\} \setminus \bigcup_{l\in[L]}\{N_{(l),l}(t) \leq n-1\}\right)\right]$$

$$\leq \sum_{n=1}^{\infty} K^2 n \mathbb{P}\left[\bigcup_{t=1}^{T}\left(\bigcup_{l\in[L]} d_{\mathrm{KL}}(\hat{\mu}_{(l),l}(t), \theta^*_{(l)}\kappa^*_l)\} > f(N_{(l),l}(t)), \bigcap_{l\in[L]}\{N_{(l),l}(t) \geq n\}\right)\right]$$

$$\leq \sum_{n=1}^{\infty} K^2 n \sum_{l\in[L]}\sum_{n_l\geq n} \mathbb{P}\left[d_{\mathrm{KL}}(\hat{\mu}^{n_l}_{(l),l}, \theta^*_{(l)}\kappa^*_l)\} > f(n_l)\right]$$

$$\leq \sum_{n=1}^{\infty} 2K^2 n \sum_{l\in[L]}\sum_{n_l\geq n} e^{-n_l f(n_l)}$$

(by Chernoff bound from both sides)

$$= O(1).$$

Taking a union bound of $(1),\ldots,(L)$ over all $L$ distinct elements among $[K]$ complete the proof. $\square$

**Lemma 17.** (Minimum divergence gap) *There exists a constant $C_{\mathrm{mgap}}$ such that*

$$\left\{N_{i,l}(t) > \alpha\sqrt{\log t}, \bigcap_{i,l}(d_{\mathrm{KL}}(\hat{\mu}_{i,l}(t), \theta^*_i\kappa^*_l) - f(N_{i,l}(t)))_+ = 0, \mathcal{W}^c(t)\right\}$$

*cannot occur for $t \geq C_{\mathrm{mgap}}$.*

*Proof of Lemma 17.* Event $\mathcal{W}^c(t)$ implies that there exists a pair $i,j \in [K]$ such that

$$\{(\theta^*_i - \hat{\theta}_i(t))(\theta^*_j - \hat{\theta}_j(t)) \leq 0\},$$

which implies $\max(|\theta^*_i - \hat{\theta}_i(t)|, |\theta^*_j - \hat{\theta}_j(t)|) > |\theta^*_i - \theta^*_j|/2$. Let $\Delta = |\theta^*_i - \theta^*_j|$. Without loss of generality we assume $|\theta^*_i - \hat{\theta}_i(t)| > \Delta/2$. Remember that $\kappa^*_1 = 1$. By using the Pinsker's inequality

$$\max(d_{\mathrm{KL}}(\hat{\mu}_{i,1}(t), \theta^*_i), d_{\mathrm{KL}}(\hat{\mu}_{i,1}(t), \hat{\theta}_i(t))) \geq \Delta^2/2 \tag{15}$$

Note that, by the definition of MLE

$$\bigcap_{i,l}\left\{(d_{\mathrm{KL}}(\hat{\mu}_{i,l}(t), \theta^*_i\kappa^*_l) - f(N_{i,l}(t)))_+ = 0\right\} \Rightarrow \bigcap_{i,l}\left\{\left(d_{\mathrm{KL}}(\hat{\mu}_{i,l}(t), \hat{\theta}_i(t)\hat{\kappa}_l(t)) - f(N_{i,l}(t))\right)_+ = 0\right\}. \tag{16}$$

Inequalities (15) and (16) do not hold simultaneously for $f(N_{i,l}(t)) < \Delta^2/2$, and thus the Lemma holds with $C_{\mathrm{mgap}} = \exp(\frac{16\gamma^4}{\Delta^8\alpha^2})$ by using the condition $N_{i,j}(t) > \alpha\sqrt{\log t}$. $\square$

*Proof of Lemma 10.* Let

$$\mathcal{H}(t) = \{(i,l) \in [K] \times [L] : (d_{\mathrm{KL}}(\hat{\mu}_{i,l}(t), \theta^*_i\kappa^*_l) - f(N_{i,l}))_+ > 0\}.$$

Let $\mathcal{G} \in 2^{[K]\times[L]} \setminus \emptyset$ be arbitrary. Note that, if

$$\left\{\log t \geq \sum_{(i,l)\in\mathcal{G}} N_{i,l}(t)\left(d_{\mathrm{KL}}(\hat{\mu}_{i,l}(t), \theta^*_i\kappa^*_l) - f(N_{i,l}(t))\right), \mathcal{G} = \mathcal{H}(t), \mathcal{P}^c(t)\right\}$$

then

$$\sum_{(i,l)\in\mathcal{G}} N_{i,l}(t)\left(d_{\mathrm{KL}}(\hat{\mu}_{i,l}(t),\theta_i^*\kappa_l^*)-f(N_{i,l}(t))\right)$$

$$\geq \inf_{\{\theta_i\}\in\mathcal{T}_{\hat{1}(t),\dots,\hat{L}(t)}^c,\{\kappa_l\}\in\mathcal{K}_{\mathrm{all}}:\forall_{l\in[L]}d_{\mathrm{KL}}(\hat{\mu}_{\hat{i}(t),l}(t),\theta_{\hat{i}(t)}^*\kappa_l^*)\leq f(N_{\hat{i}(t),l}(t))}\sum_{(i,l)\in[K]\times[L]}\left(d_{\mathrm{KL}}(\hat{\mu}_{i,l}(t),\theta_i\kappa_l)-f(N_{i,l}(t))\right),$$

which implies that at least one of the pairs $(i,l)$ in $\mathcal{G}$ is immediately put into $L_N$ to satisfy the constraints at $(\{\theta_i^*\},\{\kappa_l^*\})$. Therefore, one of the pairs in $\mathcal{G}$ is drawn within $K^2$ rounds because $|L_N|$ is at most $K^2$. By using this fact, we have

$$\sum_t \mathbf{1}\left[\mathcal{X}^c(t),\mathcal{W}^c(t),\mathcal{G}=\mathcal{H}(t),\mathcal{P}^c(t),N_{i,l}(t)=n_{i,l}\right]\leq\exp\left(\sum_{(i,l)\in\mathcal{G}}n_{i,l}\left(d_{\mathrm{KL}}(\hat{\mu}_{i,l},\theta_i^*\kappa_l^*)-f(N_{i,l}(t))\right)\right)+K^2.$$

Let $\hat{\mu}_{i,l}^n$ be the empirical estimate of $\mu_{i,l}^*=\theta_i^*\kappa_l^*$ with $n$ draws. Let $s_{i,l}=s_{i,l}(\mu_{i,l}^*)=\sup_{p\in[0,1]}d_{\mathrm{KL}}(p,\mu_{i,l}^*)<\infty$. Letting $\mathbb{P}_{i,l}(x_{i,l})=\mathbb{P}[d_{\mathrm{KL}}(\hat{\mu}_{i,j}^{n_{i,l}},\mu_{i,l}^*)\geq x_{i,l}]$, we have

$$\mathbb{E}\left[\sum_t\mathbf{1}\left[\mathcal{X}^c(t),\mathcal{W}^c(t),\mathcal{G}=\mathcal{H}(t),\bigcap_{(i,l)\in\mathcal{G}}N_{i,l}(t)=n_{i,l}\}\right]\right]$$

$$\leq\int_{\{x_{i,l}\}\in[f(n_{i,l}),s_{i,l}]^{|\mathcal{G}|}}\left(\exp\left(\sum_{(i,l)\in\mathcal{G}}n_{i,l}(x_{i,l}-f(n_{i,l}))\right)+K^2\right)\prod_{(i,l)\in\mathcal{G}}\mathrm{d}(-\mathbb{P}_{i,l}(x_{i,l}))$$

$$=K^2\prod_{(i,l)\in\mathcal{G}}s_{i,l}\mathbb{P}_{i,l}(f(n_{i,l}))+\prod_{(i,l)\in\mathcal{G}}\int_{x_{i,l}\in[f(n_{i,l}),s_{i,l}]}e^{n_{i,l}(x_{i,l}-f(n_{i,l}))}\mathrm{d}(-\mathbb{P}_{i,l}(x_{i,l}))$$

$$=K^2\prod_{(i,l)\in\mathcal{G}}s_{i,l}\mathbb{P}_{i,l}(f(n_{i,l}))$$

$$+\prod_{(i,l)\in\mathcal{G}}\left(\left[-e^{n_{i,l}(x_{i,l}-f(n_{i,l}))}\mathbb{P}_{i,l}(x_{i,l})\right]_{f(n_{i,l})}^{s_{i,l}}+\int_{x_{i,l}\in[f(n_{i,l}),s_{i,l}]}n_{i,l}e^{n_{i,l}(x_{i,l}-f(n_{i,l}))}\mathbb{P}_{i,l}(x_{i,l})\mathrm{d}x_{i,l}\right)$$

(integration by parts)

$$\leq(1+K^2)\prod_{(i,l)\in\mathcal{G}}s_{i,l}\mathbb{P}_{i,l}(f(n_{i,l}))+\prod_{(i,l)\in\mathcal{G}}\int_{x_{i,l}\in[f(n_{i,l}),s_{i,l}]}n_{i,l}e^{n_{i,l}(x_{i,l}-f(n_{i,l}))}e^{-n_{i,l}x_{i,l}}\mathrm{d}x_{i,l}$$

$$\leq(1+K^2)\prod_{(i,l)\in\mathcal{G}}s_{i,l}e^{-n_{i,l}f(n_{i,l})}+\prod_{(i,l)\in\mathcal{G}}\int_{x_{i,l}\in[f(n_{i,l}),s_{i,l}]}n_{i,l}e^{-n_{i,l}f(n_{i,l})}\mathrm{d}x_{i,l}$$

$$=(1+K^2)\prod_{(i,l)\in\mathcal{G}}s_{i,l}e^{-n_{i,l}f(n_{i,l})}+\prod_{(i,l)\in\mathcal{G}}s_{i,l}n_{i,l}e^{-n_{i,l}f(n_{i,l})}.\tag{17}$$

By summing (17) over $\{n_{i,l}\}$,

$$\sum_{t=1}^T\mathbb{P}\left[\mathcal{X}^c(t),\mathcal{W}^c(t),\mathcal{G}=\mathcal{H}(t),\mathcal{P}^c(t)\right]$$

$$\leq\sum_{\{n_{i,l}\}\in\mathbb{N}^{|\mathcal{G}|}}\cdots\sum\left((1+K^2)\prod_{(i,l)\in\mathcal{G}}s_{i,l}e^{-n_{i,l}f(n_{i,l})}+\prod_{(i,l)\in\mathcal{G}}s_{i,l}n_{i,l}e^{-n_{i,l}f(n_{i,l})}\right)$$

$$=O(1).$$

Taking a union bound over $\mathcal{G}\neq\emptyset$ yields

$$\mathbb{P}\left[\mathcal{X}^c(t),\mathcal{W}^c(t),\mathcal{H}(t)\neq\emptyset,\mathcal{P}^c(t)\right]=O(1).\tag{18}$$

We finally obtain

$$\mathbb{P}\left[\mathcal{X}^c(t),\mathcal{W}^c(t)\right]=\mathbb{P}\left[\mathcal{P}(t)\right]+\mathbb{P}\left[\mathcal{X}^c(t),\mathcal{H}(t)=\emptyset,\mathcal{W}^c(t)\right]+\mathbb{P}\left[\mathcal{X}^c(t),\mathcal{W}^c(t),\mathcal{H}(t)\neq\emptyset,\mathcal{P}^c(t)\right]=O(1),$$

where each term is bounded by Lemma 16, Lemma 17, and inequality (18). □

# J Proof of Lemma 11

Following Hogan [16], we define the continuity of a point-to-set map $\Omega : X \to 2^Y$ between metric spaces $X$ and $Y$ as follows: $\Omega$ is open at $x_0 \in X$ if $\{x^k\}$, $x^k \to x_0$, and $y_0 \in \Omega(x_0)$ imply the existence of an integer $m$ and a sequence $\{y^k\}$ such that $y^k \in \Omega(x^k)$ for $k \geq m$ and $y^k \to y_0$. $\Omega$ is closed at $x_0$ if $\{x^k\} \in X$, $x^k \to x_0$, $y^k \to y_0$ imply that $y_0 \in \Omega(x_0)$. Moreover, $\Omega$ is continuous at $x_0$ if it is closed and open at $x_0$.

Let $\mathcal{Q}^{\mathrm{L}} = \{\{q_{i,l}\}_{(i,l) \in [K] \times [L]} \in [0, \infty)^{K \times L} : \exists_{\{q_{i,l}\}_{i \in [K], l \in [K] \setminus [L]}}, \{q_{i,l}\}_{(i,l) \in [K] \times [K]} \in \mathcal{Q}\}$ be a restriction of $\mathcal{Q}$ into $K \times L$ dimension. Note that the convexity of $\mathcal{Q}^{\mathrm{L}}$ follows from the convexity of $\mathcal{Q}$. Let a set of feasible solutions restricted to $[K] \times [L]$ space be

$$\mathcal{R}^{\mathrm{L}}_{(1),\ldots,(L)}(\{\mu_{i,l}\}, \{\theta_i\}, \{\kappa_l\}, \{\delta_{i,l}\}) = \Bigg\{ \{q_{i,l}\} \in \mathcal{Q}^{\mathrm{L}} :$$

$$\inf_{\{\theta_i'\} \in \mathcal{T}^c_{(1),\ldots,(L)}, \{\kappa_l'\} \in \mathcal{K}_{\mathrm{all}}: \forall_{l \in [L]} d_{\mathrm{KL}}(\mu_{(l),l}, \theta_{(l)}' \kappa_l') \leq \delta_{i,l}} \sum_{(i,l) \in [K] \times [L]: i \neq (l)} q_{i,l} d_{\mathrm{KL}}\left(\mu_{i,l}, \theta_i' \kappa_l'\right) \geq 1 \Bigg\}. \quad (19)$$

The set of the optimal solutions $\mathcal{R}^{*,\mathrm{L}}_{(1),\ldots,(L)}(\cdot)$ in $K \times L$ dimensions are defined in accordance with $\mathcal{R}^{\mathrm{L}}_{(1),\ldots,(L)}(\cdot)$, that is,

$$\mathcal{R}^{*,\mathrm{L}}_{(1),\ldots,(L)}(\{\mu_{i,l}\}, \{\theta_i\}, \{\kappa_l\}, \{\delta_{i,l}\}) = \Bigg\{ \{q_{i,l}\} \in \mathcal{Q}^{\mathrm{L}} : \exists_{\{q_{i,l}\}_{i \in [K], l \in [K] \setminus [L]}: \{q_{i,l}\}_{(i,l) \in [K] \times [K]} \in \mathcal{Q}},$$

$$\sum_{(i,l) \in [K] \times [L]} \Delta_{i,l} q_{i,l} = C^{*,\mathrm{H}}_{(1),\ldots,(L)}(\{\mu_{i,l}\}, \{\theta_i\}, \{\kappa_l\}, \{\delta_{i,l}\}) \Bigg\}.$$

Let the norms on $\{\theta_{i,l}\}$ and $\{q_{i,l}\}$ be $|\{\theta_{i,l}\}| = \sum_{i,l} |\theta_{i,l}|$ and $|\{q_{i,l}\}| = \sum_{i,l} |q_{i,l}|$, respectively. In the following, we show the following lemma:

**Lemma 18.** (The continuity of the solution function) *The point-to-set map* $\mathcal{R}^{*,\mathrm{L}}_{(1),\ldots,(L)}(\{\mu_{i,l}\}, \{\theta_i\}, \{\kappa_l\}, \{\delta_{i,l}\})$ *is continuous at* $(\{\mu_{i,l}\}, \{\theta_i\}, \{\kappa_l\}, \{\delta_{i,l}\}) = (\{\mu^*_{i,l}\}, \{\theta^*_i\}, \{\kappa^*_i\}, \{0\})$.

The continuity and the uniqueness of the optimal solution function $\mathcal{R}^{*,\mathrm{L}}_{(1),\ldots,(L)}(\{\mu_{i,l}\}, \{\theta_i\}, \{\kappa_l\}, \{\delta_{i,l}\})$ implies that all solutions of $\mathcal{R}^{*,\mathrm{L}}_{(1),\ldots,(L)}(\{\mu_{i,l}\}, \{\theta_i\}, \{\kappa_l\}, \{\delta_{i,l}\})$ approach $\mathcal{R}^{*,\mathrm{L}}_{(1),\ldots,(L)}(\{\mu^*_{i,l}\}, \{\theta^*_i\}, \{\kappa^*_i\}, \{0\})$ when $(\{\mu_{i,l}\}, \{\theta_i\}, \{\kappa_l\}, \{\delta_{i,l}\})$ is sufficiently close to $(\{\mu^*_{i,l}\}, \{\theta^*_i\}, \{\kappa^*_i\}, \{0\})$. To prove Lemma 18, we first restate the following three Lemmas of Hogan [16]:

**Lemma 19.** (Hogan [16, Theorem 10]) *Let $g$ be a real-valued function on $X \times Y$, and $P(x) = \{y \in Y : g(x, y) \leq 0\}$ be a map of feasible solutions. If $g$ is continuous on $x_0 \times Y$, then $P$ is closed at $x_0$.*

**Lemma 20.** (Hogan [16, Theorem 13]) *Let $I(x) = \{y \in Y : g(x, y) < 0\} \subset P(x)$. If $Y$ is convex and normed, if $g$ is continuous on $x_0 \times P(x_0)$ and $P(x_0) \subset cl(I(x_0))$, then $P$ is open at $x_0$, where $cl(\cdot)$ is a closure operator.*

**Lemma 21.** (Hogan [16, Corollary 8.1]) *Let $\Omega : X \to 2^Y$ be a point-to-set map and $M(x) = \{y \in \Omega(x) : \sup_{y' \in \Omega(x)} f(x, y') = f(x, y)\}$ be an optimal solution function of some real-valued function $f$ on $X \times Y$. Suppose $\Omega$ is continuous at $x_0$, $f$ is continuous on $x_0 \times \Omega(x_0)$, $M$ is non-empty and uniformly compact near $x_0$, and $M(x_0)$ is unique. Then, $M$ is continuous at $x_0$.*

*Proof of Lemma 18.* We first show the continuity of the feasible solution function $\mathcal{R}^{\mathrm{L}}_{(1),\ldots,(L)}(\cdot)$ at $(\{\mu^*_{i,l}\}, \{\theta^*_i\}, \{\kappa^*_i\}, \{0\})$. Applying Lemma 19 for $P = \mathcal{R}^{\mathrm{L}}_{(1)\ldots(L)}$, $x_0 = (\{\mu^*_{i,l}\}, \{\theta^*_i\}, \{\kappa^*_i\}, \{0\})$

and

$$g\Big(\big(\{\mu_{i,l}\}, \{\theta_i\}, \{\kappa_l\}, \{\delta_{i,l}\}\big), \{q_{i,l}\}\Big)$$

$$= \sup_{\{\theta_i'\} \in \mathcal{T}_{1,\dots,L}^c, \{\kappa_l'\} \in \mathcal{K}_{\mathrm{all}}: \forall l \in [L] \, d_{\mathrm{KL}}(\mu_{(l),l}, \theta_{(l)}' \kappa_l') \le \delta_{i,l}} \left(1 - \sum_{(i,l) \in [K] \times [L]: i \ne (l)} q_{i,l} d_{\mathrm{KL}}(\mu_{i,l}, \theta_i' \kappa_l')\right)$$

yields the closedness of $\mathcal{R}_{(1)\dots(L)}^{\mathrm{L}}$ at $(\{\mu_{i,l}^*\}, \{\theta_i^*\}, \{\kappa_i^*\}, \{0\})$, where the continuity of $g$ follows from the uniform continuity of the KL divergence in the region where $\theta_i' \kappa_l'$ is sufficiently far from $\{0, 1\}$. Moreover, let $\{\mu_{i,l}\}, \{\theta_i\}, \{\kappa_l\}, \{\delta_{i,l}\}, \{q_{i,l}\}$ be such that $0 = g((\{\mu_{i,l}\}, \{\theta_i\}, \{\kappa_l\}, \{\delta_{i,l}\}), \{q_{i,l}\})$. Then $\{(1-\epsilon)q_{i,l}\} \in \mathcal{Q}^{\mathrm{L}}$ and $g((\{\mu_{i,l}\}, \{\theta_i\}, \{\kappa_l\}, \{\delta_{i,l}\}), \{(1-\epsilon)q_{i,l}\}) = \epsilon$ for any $\epsilon \in [0, 1]$, which implies that $\{q_{i,l}\}$ is in $cl(I((\{\mu_{i,l}\}, \{\theta_i\}, \{\kappa_l\}, \{\delta_{i,l}\})))$. By the fact above and the continuity of $g$, applying Lemma 20 to the same $P, x_0, g$ yields the openness of $\mathcal{R}_{(1)\dots(L)}^{\mathrm{L}}$ at $(\{\mu_{i,l}^*\}, \{\theta_i^*\}, \{\kappa_i^*\}, \{0\})$. The continuity of $\mathcal{R}_{(1)\dots(L)}^{\mathrm{L}}$ follows from its closedness and the openness.

Finally, by using the continuity of $\mathcal{R}_{(1)\dots(L)}^{\mathrm{L}}$ and $C_{(1),\dots,(L)}^*$, and uniform compactness and uniqueness of $\mathcal{R}_{(1),\dots,(L)}^{*,\mathrm{L}}$ at $(\{\mu_{i,l}^*\}, \{\theta_i^*\}, \{\kappa_i^*\}, \{0\})$, applying Lemma 21 to $M = \mathcal{R}_{(1),\dots,(L)}^{*,\mathrm{L}}, \Omega = \mathcal{R}_{(1)\dots(L)}^{\mathrm{L}}$, and $f = \mathcal{R}_{(1),\dots,(L)}^{*,\mathrm{L}}$ yields the continuity of $\mathcal{R}_{(1),\dots,(L)}^{*,\mathrm{L}}$ at $(\{\mu_{i,l}^*\}, \{\theta_i^*\}, \{\kappa_i^*\}, \{0\})$. $\qquad \square$

*Proof of Lemma 11.* By using the continuity of $\mathcal{R}_{1,\dots,L}^{*,\mathrm{L}}(\{\mu_{i,l}^*\}, \{\theta_i^*\}, \{\kappa_i^*\}, \{0\})$ (Lemma 18) and the uniqueness of $\mathcal{R}_{(1),\dots,(L)}^{*,\mathrm{L}}(\{\mu_{i,l}^*\}, \{\theta_i^*\}, \{\kappa_i^*\}, \{0\})$, there exists $\epsilon(\delta)$ such that $\epsilon \to 0$ as $\delta \to +0$ and

$$\sum_{t=1}^{T} \mathbf{1}[\mathcal{Y}_{i,l}(t), \mathcal{Z}_\delta(t)] \le \sum_{n=1}^{T} \mathbf{1}\left[\bigcup_{t=1}^{T} \{\mathcal{Y}_{i,j}(t), \mathcal{Z}_\delta(t), N_{i,l}(t) = n\}\right]$$

$$\le \sum_{n=1}^{T} \mathbf{1}\left[\bigcup_{t=1}^{T} \left\{n/\log t \le (1+\epsilon(\delta))\left(\mathcal{R}_{(1),\dots,(L)}^{*,\mathrm{L}}(\{\mu_{i,l}^*\}, \{\theta_i^*\}, \{\kappa_i^*\}, \{0\})\right)_{i,l}\right\}\right]$$

$$\le (1+\epsilon(\delta))\left(\mathcal{R}_{(1),\dots,(L)}^{*,\mathrm{L}}(\{\mu_{i,l}^*\}, \{\theta_i^*\}, \{\kappa_i^*\}, \{0\})\right)_{i,l} \log T + 1,$$

where $\left(\mathcal{R}_{(1),\dots,(L)}^{*,\mathrm{L}}(\{\mu_{i,l}^*\}, \{\theta_i^*\}, \{\kappa_i^*\}, \{0\})\right)_{i,l}$ is the corresponding $(i, l)$ entry, which completes the proof. $\qquad \square$

## K  Proof of Lemma 12

**Lemma 22.** *Let $C_2 > 0$ be arbitrary. Assume that $|\hat{\theta}_i(t) - \hat{\theta}_j(t)| \ge C_2$ for any $i, j \in [K] \times [K]$ and $|\hat{\kappa}_l(t) - \hat{\kappa}_m(t)| \ge C_2$ for any $l \ne m \in [L]$ hold. Moreover, assume that $f(N_{i,l}(t)) \le C_2^2/4$ holds for any $(i, l) \in [K] \times [L]$. Then, for any optimal solution, the following inequality holds for any $(i, l) \notin \{(\hat{1}(t), 1), (\hat{2}(t), 2), \dots, (\hat{L}(t), L)\}$:*

$$(\mathcal{R}_{\hat{1}(t),\dots,\hat{L}(t)}^{*,\mathrm{L}})_{i,l}(\{\hat{\mu}_{i,l}(t)\}, \{\hat{\theta}_i(t)\}, \{\hat{\kappa}_i(t)\}, \{f(N_{i,l}(t))\}) \le \frac{4LK}{C_2^4}.$$

*Proof of Lemma 22.* If $\{\theta_i'\} \in \mathcal{T}_{\hat{1}(t),\dots,\hat{L}(t)}^c$, then there exists $i', j'$ such that $(\theta_{i'}' - \theta_{j'}')(\hat{\theta}_i(t) - \hat{\theta}_j(t)) \le 0$, which implies $\max(|\theta_{i'}' - \hat{\theta}_{i'}(t)|, |\theta_{j'}' - \hat{\theta}_{j'}(t)|) \ge C_2/2$. Without loss of generality let

$|\theta'_{i'} - \hat{\theta}_{i'}(t)| \geq C_2/2$. Then, for any $\{\theta'_i\}, \{\kappa'_l\}$,

$$\sum_{i,l} q_{i,l} \left( d_{\mathrm{KL}}(\hat{\mu}_{i,l}(t), \theta'_i \kappa'_l) - f(N_{i,l}(t)) \right)_+$$

$$\geq q_{i',1} \left( d_{\mathrm{KL}}(\hat{\theta}_{i'}(t), \theta'_{i'}) - f(N_{i',1}(t)) \right)_+$$

$$\geq q_{i',1} \left( 2(\hat{\theta}_{i'}(t) - \theta'_{i'})^2 - f(N_{i',1}(t)) \right)_+$$

(by Pinsker's inequality)

$$\geq q_{i',1} \left( C_2^2/2 - C_2^2/4 \right)_+$$

$$= q_{i',1} C_2^2/4,$$

which implies that

$$\left\{ q_{i,l} = \frac{4}{C_2^2} \right\}_{(i,l) \in [K] \times [L]} \in \mathcal{R}^{\mathrm{L}}_{\hat{1}(t), \dots, \hat{L}(t)}(\{\hat{\mu}_{i,l}(t)\}, \{\hat{\theta}_i(t)\}, \{\hat{\kappa}_i(t)\}, \{f(N_{i,l}(t))\}).$$

The estimated regret on the basis of $\{\hat{\theta}_i(t)\}, \{\hat{\kappa}_l(t)\}$ increases at least $\min_{i \neq j} \min_{l \neq m} |\hat{\theta}_i(t) - \hat{\theta}_j(t)||\hat{\kappa}_l(t) - \hat{\kappa}_m(t)| \geq C_2^2$ when we draw arms that are not $\hat{1}(t), \dots \hat{L}(t)$. Therefore, $q_{i,l}$ for $i \neq \hat{l}(t)$ is bounded by $\frac{4LK}{C_2^4}$. $\qquad\square$

Let

$$\mathcal{A}(t) = \bigcap_{i,l} \{ d_{\mathrm{KL}}(\hat{\mu}_{i,l}(t), \theta_i^* \kappa_l^*) \leq f(N_{i,l}(t)) \} .$$

Then, the following lemma holds.

**Lemma 23.** *For sufficiently small $\delta > 0$, we have*

$$\sum_{t=1}^T \mathbb{P} \left[ N_{i,l}(t) > \alpha \sqrt{\log t}, \mathcal{Z}_\delta^c(t), \mathcal{A}(t) \right] = O(1).$$

*Proof of Lemma 23.* Let

$$\mathcal{B}_\delta(t) = \bigcap_{i,l} \left\{ f(N_{i,l}(t)) < \delta^2 \right\} .$$

Then,

$$\sum_{t=1}^T \mathbb{P} \left[ \mathcal{X}^c(t), \mathcal{B}_\delta^c(t) \right]$$

$$\leq \sum_{(i,l) \in [K] \times [L]} \sum_{t=1}^T \mathbb{P} \left[ N_{i,j}(t) \geq \alpha \sqrt{\log t}, f(N_{i,l}(t)) > \delta^2 \right]$$

$$\leq \sum_{(i,l) \in [K] \times [L]} \sum_{t=1}^T \mathbb{P} \left[ \gamma \frac{1}{\alpha^{1/2}(\log t)^{1/4}} > \delta^2 \right]$$

$$\leq K^2 e^{\left( \frac{\gamma^4}{\alpha^2 \delta^8} \right)} = O(1). \tag{20}$$

Moreover, $\mathcal{A}(t)$ implies

$$\bigcap_{i,l} \left\{ d_{\mathrm{KL}}(\hat{\mu}_{i,l}(t), \hat{\theta}_i(t)\hat{\kappa}_l(t)) \leq f(N_{i,l}(t)) \right\} ,$$

and by using the Pinsker's inequality we have

$$|\hat{\mu}_{i,l}(t) - \hat{\theta}_i(t)\hat{\kappa}_l(t)| \leq \frac{1}{2} \sqrt{f(N_{i,l}(t))}$$

$$|\hat{\mu}_{i,l}(t) - \theta_i^* \kappa_l^*| \leq \frac{1}{2} \sqrt{f(N_{i,l}(t))},$$

for any $(i, l) \in [K] \times [L]$, which implies

$$|\hat{\theta}_i(t)\hat{\kappa}_l(t) - \theta_i^* \kappa_l^*| \leq \sqrt{f(N_{i,l}(t))}$$

for any $(i, l) \in [K] \times [L]$. Therefore,

$$\sum_{t=1}^{T} \mathbb{P}\left[\mathcal{X}^c(t), \mathcal{Z}_\delta^c(t), \mathcal{A}(t)\right]$$

$$\leq \sum_{t=1}^{T} \mathbb{P}\left[\mathcal{X}^c(t), \mathcal{B}_\delta^c(t)\right] + \sum_{t=1}^{T} \mathbb{P}\left[\mathcal{X}^c(t), \mathcal{B}_\delta(t), \mathcal{Z}_\delta^c(t), \mathcal{A}(t)\right]$$

$$\leq O(1) + \mathbb{P}\left[\mathcal{Z}_\delta^c(t), \bigcap_{(i,l)\in[K]\times[L]} \{f(N_{i,l}(t)) \leq \delta^2, |\hat{\theta}_i(t)\hat{\kappa}_l(t) - \theta_i^* \kappa_l^*| \leq \sqrt{f(N_{i,l}(t))}\}\right]$$

(by inequality (20))

$$\leq O(1) + \mathbb{P}\left[\mathcal{Z}_\delta^c(t), \bigcap_{(i,l)\in[K]\times[L]} \{|\hat{\theta}_i(t)\hat{\kappa}_l(t) - \theta_i^* \kappa_l^*| \leq \delta\}\right] = O(1) + 0.$$

$\square$

**Lemma 24.** *The following inequality holds:*

$$\sum_{t=1}^{T} \mathbf{1}\left[\mathcal{X}^c(t), \mathcal{B}_{(\beta/(\log\log t))/4}^c(t)\right] = O(1).$$

*Proof of Lemma 24.* We have

$$\sum_{(i,l)\in[K]\times[L]} \sum_{t=1}^{T} \mathbf{1}\left[\mathcal{X}^c(t), \mathcal{B}_{(\beta/(\log\log t))/4}^c(t)\right] \leq \sum_{(i,l)\in[K]\times[L]} \sum_{t=1}^{T} \mathbf{1}\left[\frac{\gamma}{\sqrt{\alpha}\sqrt{\log t}} > \frac{\beta^2}{4(\log\log t)}\right]$$

$$\leq \sum_{(i,l)\in[K]\times[L]} \sum_{t=1}^{T} \mathbf{1}\left[\frac{\log t}{(\log\log t)^4} < \frac{2^8\gamma^4}{\alpha^2\beta^4}\right] = O(1).$$

$$\left(\text{by } \frac{\log t}{(\log\log t)^8} \to \infty \text{ as } t \to \infty\right)$$

$\square$

*Proof of Lemma 12.* We have

$$\sum_{t=1}^{T} \mathbb{P}\left[\mathcal{X}^c(t), \mathcal{Y}_{i,l}(t), \mathcal{Z}_\delta^c(t)\right]$$

$$\leq \sum_{t=1}^{T} \mathbb{P}\left[\mathcal{X}^c(t) \bigcap_{(i',l')\in[K]\times[L]} N_{i',l'}(t) \geq (\log\log T)^{1/3}, \mathcal{Y}_{i,l}(t), \mathcal{Z}_\delta^c(t)\right]$$

$$+ \sum_{(i',l')\in[K]\times[L]} \sum_{t=1}^{T} \mathbb{P}[N_{i',l'}(t) \leq (\log\log T)^{1/3}, N_{i',l'}(t) \geq \alpha\sqrt{\log t}]$$

$$\leq \sum_{t=1}^{T} \mathbb{P}\left[\mathcal{X}^c(t), \bigcap_{(i',l')\in[K]\times[L]} N_{i',l'}(t) \geq (\log\log T)^{1/3}, \mathcal{Y}_{i,l}(t), \mathcal{A}^c(t)\right]$$

$$+ \sum_{(i',l')\in[K]\times[L]} \sum_{t=1}^{T} \mathbb{P}[N_{i',l'}(t) \leq (\log\log T)^{1/3}, N_{i',l'}(t) \geq \alpha\sqrt{\log t}]$$

$$+ \sum_{t=1}^{T} \mathbb{P}\left[\mathcal{X}^c(t), \mathcal{Z}_\delta^c(t), \mathcal{A}(t)\right]. \tag{21}$$

Here, the second term of (21) is bounded as:

$$\sum_{(i',l')\in[K]\times[L]} \sum_{t=1}^{T} \mathbb{P}[N_{i',l'}(t) \leq (\log\log T)^{1/3}, N_{i',l'}(t) \geq \alpha\sqrt{\log t}] \leq K^2 e^{\alpha^{-2}(\log\log T)^{2/3}} = o(\log T).$$
$$\tag{22}$$

The third term of (21) is $O(1)$ by Lemma 23. In the following we bound the first term of (21).

$$\sum_{t=1}^{T} \mathbb{P}\left[\mathcal{X}^c(t), \bigcap_{(i',l')\in[K]\times[L]} N_{i',l'}(t) \geq (\log\log T)^{1/3}, \mathcal{Y}_{i,l}(t), \mathcal{A}^c(t)\right]$$

$$\leq \sum_{t=1}^{T} \mathbb{P}\left[\mathcal{X}^c(t), \bigcap_{(i',l')\in[K]\times[L]} N_{i',l'}(t) \geq (\log\log T)^{1/3}, \mathcal{Y}_{i,l}(t), \mathcal{A}^c(t), \mathcal{B}_{(\beta/(\log\log t))/4}(t)\right] + O(1)$$
$$\text{(by Lemma 24)}$$

$$\leq \sum_{n=1}^{T} \mathbb{P}\left[\bigcup_{t=n}^{T}\left\{\mathcal{X}^c(t), \bigcap_{(i',l')\in[K]\times[L]} N_{i',l'}(t) \geq (\log\log T)^{1/3}, \mathcal{Y}_{i,l}(t), \mathcal{A}^c(t), \mathcal{B}_{(\beta/(\log\log t))/4}(t), N_{i,l}(t) = n\right\}\right]$$
$$+ O(1)$$

$$\leq \sum_{n=1}^{\log T(4LK(\log\log T/\beta)^4/2)} \mathbb{P}\left[\bigcup_{t=1}^{T}\left\{N_{i',l'}(t) \geq (\log\log T)^{1/3}, \mathcal{A}^c(t), N_{i,l}(t) = n\right\}\right] + O(1)$$

(by the fact that $\mathcal{X}^c(t)$ combined with Lemma 22 with $C_2 = \beta/\log\log t$

$$\text{imply that } q_{i,l} \leq 4LK(\log\log T/\beta)^4 \text{ for any } (l) \neq i)$$

$$\leq e^{-\Omega((\log\log T)^{1/3})} O((\log T)(\log\log T)^4) = o(\log T), \tag{23}$$

where, in the last line we used the fact that

$$\mathbb{P}\left[\bigcup_{t=1}^{T}\left\{\mathcal{X}^c(t), \mathcal{A}^c(t), \bigcap_{(i',l')\in[K]\times[L]} N_{i',l'}(t) \geq (\log\log T)^{1/3}\right\}\right]$$

$$\leq \sum_{(i',l')\in[K]\times[L]} \sum_{n=(\log\log T)^{1/3}}^{T} \left(\mathbb{P}[|\hat{\mu}_{i',l'}^n - \mu_{i',l'}^*| > \delta] + \mathbf{1}\{\delta \geq f(\alpha\sqrt{\log t})\}\right)$$

$$\leq \sum_{(i',l')\in[K]\times[L]} \sum_{n=(\log\log T)^{1/3}}^{T} \mathbb{P}[|\hat{\mu}_{i',l'}^n - \mu_{i',l'}^*| > \delta] + O(1)$$

$$\leq \sum_{(i',l')\in[K]\times[L]} \sum_{n=(\log\log T)^{1/3}}^{T} 2e^{-2n\delta} = e^{-\Omega((\log\log T)^{1/3})} + O(1).$$

Combining (21), (22), and (23) completes the proof. □

## L Notation

Table 2 summarizes the notation appeared in this paper.

Table 2: List of notation that are used in more than one sections in this paper.

| | |
|---|---|
| $T$ | Number of rounds (Sec. 2) |
| $K$ | Number of arms (Sec. 2) |
| $L$ | Number of slots (Sec. 2) |
| $\theta_i^*$ | Parameter associated with arm $i \in [K]$ (Sec. 2) |
| $\kappa_l^*$ | Parameter associated with slot $l \in [L]$ (Sec. 2) |
| $\mu_{i,l}^*$ | $= \theta_i^* \kappa_l^*$ (Sec. 2) |
| $\Delta_{i,l}$ | $= \theta_l^* \kappa_l^* - \theta_i^* \kappa_l^*$ (Sec. 2) |
| $I(t) = (I_1(t), \dots, I_L(t))$ | List of arms selected at round $t$ (Sec. 2) |
| $N_{i,l}(t)$ | Number of rounds before $t$ at which arm $i$ was in slot $l$ (Sec. 2) |
| $\hat{\mu}_{i,l}(t)$ | Empirical mean of pair $(i,l)$ at round $t$ (Sec. 2) |
| $(1), \dots, (K)$ | A permutation of $1, \dots, K$ (Sec. 3) |
| $\mathcal{T}_{\text{all}}$ | Set of possible values of parameters on the arms (Sec. 3) |
| $\mathcal{K}_{\text{all}}$ | Set of possible values of parameters on the slots (Sec. 3) |
| $\mathcal{T}_{(1),\dots,(L)}$ | Subset of $\mathcal{T}_{\text{all}}$ such that the $i$-th best arm is $(i)$ (Sec. 3) |
| $\mathcal{T}_{(1),\dots,(L)}^c$ | $\mathcal{T}_{\text{all}} \setminus \mathcal{T}_{(1),\dots,(L)}$ (Sec. 3) |
| $d_{\text{KL}}(p, q)$ | $p \log(p/q) + (1-p) \log((1-p)/(1-q))$ (Sec. 3) |
| $\mathcal{Q}$ | Subspace of $[0,\infty)^{[K] \times [K]}$ that corresponds to the number of draws (Sec. 3) |
| $\mathcal{R}_{(1),\dots,(L)}(\{\mu_{i,l}\}, \{\theta_i\}, \{\kappa_l\})$ | Feasible solutions under parameters $\{\mu_{i,l}\}, \{\theta_i\}, \{\kappa_l\}$ and top-$L$ arms $(1), \dots, (L)$ (Sec. 3) |
| $C_{(1),\dots,(L)}^*(\{\mu_{i,l}\}, \{\theta_i\}, \{\kappa_l\})$ | Coefficient of optimal regret bound (Sec. 3) |
| $\mathcal{R}_{(1),\dots,(L)}^*(\{\mu_{i,l}\}, \{\theta_i\}, \{\kappa_l\})$ | Optimal solutions (Sec. 3) |
| $\alpha > 0$ | Parameter of PMED algorithm (Sec. 4) |
| $L_C$ | List of $L$-allocations that are drawn at current loop of PMED (Sec. 4) |
| $L_N$ | List of $L$-allocations that are drawn at next loop of PMED (Sec. 4) |
| $v_m^{\text{mod}}$ | An $L$-allocation (Sec. 4) |
| $\{\hat{\theta}_i(t)\}_{i=1}^K, \{\hat{\kappa}_l(t)\}_{l=1}^L$ | MLE. Note that $\hat{\mu}_{i,l}(t) \neq \hat{\theta}_i(t)\hat{\kappa}_l(t)$. (Sec. 4) |
| $\tilde{N}_{i,l}$ | $K \times K$ variables that correspond to the estimated amount of exploration on pair $(i,l)$ (Sec. 4) |
| $e_v$ | A permutation matrix (Sec. 4) |
| $e_{v,i,l}$ | $(i,l)$ entry of $e_v$ (Sec. 4) |
| $(\hat{1}(t), \dots, \hat{L}(t))$ | Top-$L$ arms estimated from MLE (Sec. 4) |
| $(x)_+$ | $= \max(x, 0)$ (Sec. 7) |
| $\beta, \gamma > 0$ | Parameters of PMED-Hinge algorithm (Sec. 7) |
| $f(n)$ | $= \gamma/\sqrt{n}$ (Sec. 7) |
| $\mathcal{R}_{(1),\dots,(L)}^{\text{H}}(\{\mu_{i,l}\}, \{\theta_i\}, \{\kappa_l\}, \{\delta_{i,l}\})$ | Feasible solutions of the modified objective (Sec. 7) |
| $C_{(1),\dots,(L)}^{*,\text{H}}(\{\mu_{i,l}\}, \{\theta_i\}, \{\kappa_l\}, \{\delta_{i,l}\})$ | Coefficient of optimal regret bound of modified objective (Sec. 7) |
| $\mathcal{R}_{(1),\dots,(L)}^{*,\text{H}}(\{\mu_{i,l}\}, \{\theta_i\}, \{\kappa_l\}, \{\delta_{i,l}\})$ | Optimal solutions of the modified objective (Sec. 7) |
| $\mathcal{W}(t)$ | Event that top-$L$ arms is misidentified (Sec. G) |
| $\mathcal{J}_{i,l}(t)$ | Event that pair $(i,l)$ is put into $L_N$ (Sec. G) |
| $\mathcal{X}(t)$ | Event that at least one arm is put into $L_N$ before Line 18 (Sec. G) |
| $\mathcal{Y}_{i,l}(t)$ | Eevent that pair $(i,l)$ is put into $L_N$ in Line 18 (Sec. G) |
| $\mathcal{Z}_\delta(t)$ | Event that estimator $\{\hat{\theta}_i(t), \hat{\kappa}_l(t)\}$ is $\delta$-close to the set of true values (Sec. G) |
| $R_{i,l}^*$ | $(i,l)$ entry of the optimal solution $\mathcal{R}_{1,\dots,L}^*(\{\mu_{i,l}^*\}, \{\theta_i^*\}, \{\kappa_l^*\})$ (Sec. G) |
| $\hat{\mu}_{i,l}^n$ | $\hat{\mu}_{i,l}(t)$ when $N_{i,l}(t) = n$ (Sec. G) |
| $\mathcal{C}(t)$ | Event that hinge of MLE is zero (Sec. H) |
| $\mathcal{H}(t)$ | Subset of entries $(i,l) \in [K] \times [L]$ such that hinge of true parameters is zero (Sec. I) |