[Reviews · NeurIPS 2017]

Reviewer 1



The paper investigates a multiple-play multi-armed bandit model with position bias i.e., one that involves $K$ Bernoulli bandits (with parameters $\theta_i$ \in (0,1)$ and $L$ slots with parameters $\kapa_j$ \in (0,1]$. At each point in time $t$ the system selects $L$ arms $(I_1 (t), . . . , I_L (t))$ and receives a Bernoulli reward from each arm $Ber(\theta_{I_j(t)} \kapa_j{I_j(t)})$, $j= I_1 (t), . . . , I_L (t)$. The paper derives asymptotically optimal policies in the sense of Lai & Robbins (1985). I like the model and the analysis, but I did not have time to check all the mathematics at a satisfactory level. Finally, I would like to bring to the attention of the author(s) the papers below. [1] provides the extension of the Lai Robbins (1985) work (that involves single parameter bandits) to those with multiple unknown parameters. [2] Provides asymptotically optimal polices for unknown MDPs. [1] Optimal adaptive policies for sequential allocation problems AN Burnetas, MN Katehakis - Advances in Applied Mathematics, 1996 [2] Optimal adaptive policies for Markov decision processes AN Burnetas, MN Katehakis - Mathematics of Operations Research, 1997

Reviewer 2



** Summary ** The paper tackles the learning-to-rank problem under delayed feedback that was proposed by Lagrée et al. 2016 [LVC16]. In the latter NIPS paper, the authors analyze the problem of learning how to sequentially select ordered lists of items when the position-related parameters are known. The present contribution releases this hypothesis and proposes a way to jointly learn the best allocation strategy with respect to unknown item and position parameters. They first prove a problem dependent lower bound that comes as the solution of a constraint optimization problem. Then, they make use of this result to design an algorithm that uses MLE of the parameters as plug-ins to solve the same optimization problem and obtain allocation rules. Experiments show the behavior of their algorithm PMED on simulated and pseudo-real data and an analysis of a slightly modified version of the algorithm is provided in Section 7. ** General comment ** This is a good paper overall and I would definitely recommend accepting it. I have a few questions though (see below) but I think the way the problem is posed is very relevant and interesting. Even though the authors build on the model from [LVC16], they clearly depart from the latter work and propose an original approach. ** Questions and comments ** - I am first surprised by the way the *asymptotic* results are stated. Lemma 1, Theorem 2 and Lemma 6 are asymptotic bounds but it is not very clear in the way they are stated I think. It would be more explicit by using either a ratio "\lim_T\to \infty f(T)/log(T) \geq ore \leq ..." or to add "There exist a positive M such that for all T\geq M , ..." in the statement of the result. Or maybe I missed something and I'd be happy to read your answer. - Your algorithm is based on optimization routines. You comment on the computational complexity of the methods you use but we don't know how it translate on computation time. I know that this is a relative metric that depends on hardware and implementation but I would be curious if you could give some insight on your experiments and on how 'scalable' is the proposed approach. ** Minor remarks ** - l94-95 in the definition of \Delta and the regret, the parameters should be ^* - Maybe you could split equation (1) after the "=" sign in order not to cut the optimization problem itself, for readability - The title of the paper is long and a bit close to the one of [LVC16], I'd suggest to maybe find another, simpler one. %%% After Rebuttal %% I read the rebuttal and I thank the authors for their answers to my questions and for clarifying many points after Reviewer 2's concerns. I believe this is a strong paper and I would push for acceptance.

Reviewer 3



#Summary This submission studies a multi-armed bandit problem, PBMU. In this problem, there are L slots (to place ads) and K possible ads. At each round, the player need to choose L from the K possible ads and place them into the L slots. The reward of placing the i-th ad into the l-th slot is i.i.d. drawn from a Bernoulli distribution Ber(\theta_i \kappa_l), where \theta_i is the (unknown) quality of the i-th ad and \kappa_l is the (also unknown) position bias of the l-th slot. The authors derived a regret lower bound and proposed two algorithms for PBMU. The second algorithm has a matching regret bound. The authors also discussed how to solve the related optimization problems appeared in the algorithms. The authors give a complete characterization of the hardness of PMMAB in this paper. Their main contributions are: 1.A regret lower bound, which states that the expected regret of a strongly consistent algorithm for PMMAB is lower bounded by ClogT-o(log T). Where C is a parameter determined by every arm's mean profile and slot's position bias. 2.An algorithm for PMMAB. The authors called it Permutation Minimum Empirical Divergence(PMED) which can be compared with the Deterministic Minimum Empirical Divergence (DMED) algorithm for SMAB (see Honda and Takemura COLT2010). They give an asymptotically upper bound of a modification of PMED which is Clog T + o(log T). From the technical point, I think the regret lower bound in this paper is a standard, which follows some techniques from Honda and Takemura COLT2010. In their upper bound analysis, the authors solve some new difficulties. In PMED algorithm, they compute Maximum Likelihood Estimator and use it to compute a K times K matrix which optimizes a corresponding optimization problem in the lower bound analysis. The authors then convert the matrix into a convex combination of permutation matrix (due to Birkhoff–von Neumann Theorem). Though this idea is natural, the analysis is not straightforward. The authors analyze a modification of PMED in which they add additional terms to modify the optimization problems. They obtain a corresponding upper bound for this modified algorithm. Experimental results are attached. #Treatment of related work In the introduction part, the authors introduced some other related models and compared them with their new model. They also gave convincing reasons why this new model deserves study. To the best of my knowledge, their model is novel. However, the author did not make much effort in comparing their techniques with previous (closely related) results. This made it difficult to evaluate the technical strength of their results. #Presentation The presentation of the paper still needs some work. In Section 3, the authors defined a set \mathcal{Q}. However, it is dubious why they need to define this quality when deriving the regret lower bound (the definition of this quality is not furthered used in Section 3 and related parts in the appendix). On the other hand, the authors did use several properties of this quantity in Section 4 (to run Algorithm 2). Thus, I advise the authors to place the definition of \mathcal{Q} into Section 4 in order to eliminate unnecessary confusion and change the claim of the lower bound correspondingly. Line 16-20 of Algorithm 1 (PMED and PMED-Hinge) is difficult to follow. To my understand, \hat{N}_{i, l} is the number of necessary drawings for the pair (i, l). The algorithm then decomposes the matrix \hat{N} into a linear combination of permutation matrices and pull the arms according to the permutation matrices and the coefficients. When will c_v^{aff} be smaller than c_v^{req}? Shouldn't we pull (v_1, v_2, \ldots, v_L) for c_v^{req} times (instead of once)? Should c_v^{req} be rounded up to an integer? There are some other typos/minor issues. See the list below. 1. Line 57-58: Non-convex optimization appears in your algorithm does not necessarily mean that PBMU intrinsically requires non-convex optimization. 2. Line 63-65: Define K, or remove some details. 3. Line 86-87: at round t -> of the first t rounds. 4. Line 102-105: According to the definition, (1), (2), \ldots, (L) is a permutation of {1, 2, 3, \ldots, L}, which is not what you want. 5. Line 115-117: The fact that d_{KL} is non-convex does not necessarily make PBMU intrinsically difficult. 6. Line 118, the definition of \mathcal{Q}: What do you mean by \sum (q_{i, l} = q_{i + 1, l})? 7. Line 122: (1) is not am inequality. Do you want to refer to the Inequality in Lemma 1? "the drawing each pair (i, l)", remove "the". 8. Line 124-127: Please elaborate on the reason why the constraint \theta_i?\kappa_i?= \theta_i \kappa_i is necessary. The argument given here is not easy to follow. 9. Line 138: is based on -> is closed related to 10. Line 141: what do you mean by "clarify"? 11. Algorithm 1, Line 12: Explicitly define \hat{1}(t), \ldots, \hat{L}(t). Currently it is only implicitly defined in Line 160. 12. Algorithm 1, Line 14: "where e_v for each v is a permutation matrix" -> "where for each v, e_v is a permutation matrix". 13. Line 172-174: Explicitly define the naive greedy algorithm. 14. Line 343: native->naive. 15. Algorithm 1, Line 17: \max_{c \ge 0} ... \ge 0 -> \max_{c \ge 0} s.t. ... \ge 0. ------------------- The rebuttal addressed most of my major questions. Score changed to 7.